# Light-responsive MXenegel via interfacial host-guest supramolecular bridging

Yu-Liang Lin[1], Sheng Zheng[1], Chun-Chi Chang[1], Lin-Ruei Lee[1] & Jiun-Tai Chen[1,2] ✉

Living in the global-changing era, intelligent and eco-friendly electronic components that can sense the environment and recycle or reprogram when needed are essential for sustainable development. Compared with solid-state electronics, composite hydrogels with multi-functionalities are promising candidates. By bridging the self-assembly of azobenzene-containing supramolecular complexes and MXene nanosheets, we fabricate a MXene-based composite gel, namely MXenegel, with reversible photo-modulated phase behavior. The MXenegel can undergo reversible liquefication and solidification under UV and visible light irradiations, respectively, while maintaining its conductive nature unchanged, which can be integrated into traditional solid-state circuits. The strategy presented in this work provides an example of light-responsive conducting material via supramolecular bridging and demonstrates an exciting platform for functional soft electronics.

Solid-state materials, such as metallic wires, silicon wafers, and ceramics, have dominated our daily-life electronics for decades. The rigidity and conductive nature of these traditional components bring them the ability to integrate into a piece of printed circuit board with reliable stability. The mismatch between dynamic-fluctuated substrates and passive solid-state electronics, however, is the major issue in developing state-of-the-art technology; rigid electronics are often incompatible with soft human skin and unable to give responses according to their surroundings[1–7]. Moreover, the Sustainable Development Goals (SDGs) recently announced by the United Nations Development Programme (UNDP) have aroused ever-growing concerns about reducing toxic waste and extracting unbroken parts from discarded electronics for full-cycle reuse[8,9]. Undoubtedly, we need to reflect on whether the electronics we use are smart and eco-friendly enough for sustainable purposes[10–12]. To address these issues, it is desired to develop a new charge transporting pathway to replace traditional passive components that can sense the environmental change, while being able to reprogram or recycle them instead of rebuilding new ones[3,5–7,13–15].

Here, we turn our focus on investigating conductive composite gels with multifunctionalities. By introducing inorganic dopants into organic matrix, not only the conductivity of the gel can be enhanced but also desired properties can be incorporated, e.g., thermal responsive, light responsive, self-healing, etc[16–18]. In terms of conductive dopants, graphene, carbon nanotubes, and metallic nanoparticles have been widely explored[19–21]. For example, Pedersen et al. successfully fabricated a hydrogel network based on poly(vinyl alcohol) as matrix base and gold nanoparticles as heat generators[22]; this composite hydrogel was able to undergo a liquefication process within seconds under near-infrared (NIR) laser irradiation, releasing the incorporated cargo. Ha et al. demonstrated an electro-responsive hydrogel-based microfluidic actuator[23]; the composite hydrogel consisted of silver nanowires and biocompatible collagen I, which can realize electro-responsive nanomaterial delivery and photothermal therapy applications. Zhang et al. described conductive supramolecular hydrogels composed of modified β-cyclodextrin hosts, adamantane guests, and graphene oxide[24]; this supramolecular hydrogel possessed human-tissue-like conductivity and self-healing property that can be used for wound covering dressing. There are still numerous composite gels with diverse functionalities reported but it is still difficult to comprehensively address the issues mentioned above while maintaining acceptable conductivity[25,26]. In addition, the compatibility between synthetic organic molecules with inorganic additives remains challenging.

The transition metal carbides, nitrides, and carbonitrides, which are often referred to as MXenes, are excellent candidates for

[1]Department of Applied Chemistry, National Yang Ming Chiao Tung University, Hsinchu 300093, Taiwan. [2]Center for Emergent Functional Matter Science, National Yang Ming Chiao Tung Universiy, Hsinchu 300093, Taiwan. ✉e-mail: jtchen@nycu.edu.tw

composite hydrogels and a class of rising stars in the 2D material family[27]. MXenes have the chemical formula of $M_{n+1}X_nT_x$ ($n = 1-4$), where M is early transition metals and X is carbon and/or nitrogen; $T_x$ denotes a group of surface terminations, which are -OH, -F, and -O, generated from the selectively chemical etching process[28]. The $Ti_3C_2T_x$ MXene possesses a high electrical conductivity of 10000–20000 S/cm and a high breakdown current[29]. Also, the exceptional hydrophilicity and rich surface termination density make MXene nanosheets readily dispersible in various solvents and bring great versatility in different applications. In this work, we explore an approach where the azobenzene-containing supramolecular inclusions are intercalated in the MXene nanosheets to form a light-responsive conductive gel. The synthesized azobenzene moieties are first included in the cavities of alpha-cyclodextrin (αCD) hosts to form supramolecular inclusions and self-assemble to a lamella structure in between the MXene layers; meanwhile, the positively charged heads of the inclusions interact with the negatively charged surfaces of the MXene nanosheets. The resulting composite gel, namely MXenegel, possesses electrical conductance and photo-modulated reversible phase transition behaviors. To show the feasibility and applicability of the MXenegel, we also demonstrate the MXenegel wire hand-written by a Chinese brush and the light-responsive circuit composed of a MXenegel switch and solid-state electronics.

## Results

### Conceptual design and physicochemical characterization of MXenegel

Figure 1 displays the MXenegel formation and the intercalation of the inclusion complexes, illustrating the main concept of this work. The MXenegel is prepared by weighing a desired amount of azobenzene derivative (AzoC6), αCD, and MXene into the aqueous solution, in which the molar ratio of AzoC6 and αCD is fixed at 1:2[30]. As shown in Fig. 1a, the hydrophobic end of AzoC6, including azobenzene and alkyl chain spacer, can be included into the αCD cavity. By incubating at room temperature, the inclusion of AzoC6 and αCD, which is denoted as AzoC6@2αCD, can be formed. Meanwhile, the positively charged ends of αCD can interact with the negatively charged surfaces of MXene nanosheets via Coulombic electrostatic interactions. The unique supramolecular structure of AzoC6@2αCD make the MXenegel different from those of the MXene-based materials prepared by conventional methods[31]. As illustrated in Fig. 1b, the MXenegel can undergo a gel-to-sol transition upon the irradiation of UV light or thermal treatment, which is originated by the disassembly of the AzoC6@2αCD inclusions intercalated between the MXene layers. The sol-state MXenegel can be reformed back to a gel-like structure under the irradiation of visible light or by keeping in a dark room at room temperature, indicating that such a phase transition is reversible.

The MXene nanosheets used in this work are prepared by a less aggressive etching method, namely the minimally intensive layer delamination (MILD) method, to minimize the MXene defects, as illustrated in Fig. 2a[32]. As shown in Fig. 2c, the SEM image of the MAX precursor contains unremoved Al atoms in the layer structure. After the MILD selective etching process, the as-prepared multi-layered $Ti_3C_2T_x$ MXene clay is delaminated by a sonication bath (Fig. 2d). The delamination process does not require the addition of surfactants or stabilizers, such as DMSO and TMAOH[27]. The hydrophilicity nature of the MXene nanosheets leads to a stable suspension in aqueous solution. Characterized by DLS, the average hydrodynamic diameter of the MXene nanosheets used in this work is ca. 838 nm (Fig. 2e). It is worth mentioning that the surfaces of the MXene nanosheets possess negative charges. Therefore, through the Coulombic electrostatic interactions, MXene can form a hybrid system with other positively charged organic molecules or ligands that contain specific functionalities[33].

Figure 2b displays the synthetic scheme of the catanionic azobenzene derivative (AzoC6). Without any functional group directly attached to the azobenzene, the absorption peaks of AzoC6 photoisomerization are similar to the native 4-(phenyldiazenyl)phenol. In Fig. 2f, the UV−vis spectrum of the AzoC6@2αCD complex shows the characteristic absorption bands during the photoisomerization. The maximum at 343 nm and the relatively weak band at 430 nm are corresponding to the transitions of $\pi - \pi^*$ and $n - \pi^*$ of trans-AzoC6, respectively. The maxima of the AzoC6@2αCD sample are identical to the pristine AzoC6 in water (Supplementary Fig. 7), indicating that the formation of AzoC6@2αCD inclusions does not cause the shifting of the absorption bands. After the conversion into the cis-form, because of their geometrical differences, the $\pi - \pi^*$ transition blue-shifts to 323 nm and the $n - \pi^*$ slightly red-shifts to 432 nm. By irradiating with UV and visible lights, it is shown that the synthesized AzoC6 can reversibly undergo the trans-to-cis and cis-to-trans isomerizations (Fig. 2g), respectively. After a cycle of UV and visible light irradiations, the trans-isomer ratio decreases to 79% and remains at similar values after continuous cyclic irradiations. It should also be noted that, according to our previous work, the inclusion formation and the self-assembly of the AzoC6@2αCD complexes are much slower than the isomerization of Azo moieties[34]. Therefore, a longer irradiation time for the visible light is applied to ensure the successful inclusion formation. To further confirm the formation of AzoC6@2αCD complexes, we conduct thermal analyses by DSC. Figure 2h shows the DSC heating profiles of the AzoC6, αCD, and the vacuum-dried compound of the AzoC6/αCD mixture (1:2 molar ratio). The thermal curve of AzoC6 powder displays a sharp endothermic peak at 148 °C, which corresponds to its melting peak. The DSC curve of αCD shows three broad endothermic peaks at 79, 115, and 139 °C. These peaks are mainly attributed to the water loss from αCD crystals[35,36]. Also, a broad endothermic peak at 80−95 °C is recorded for the AzoC6/2αCD mixture as a consequence of water loss from αCD. The entire heating trace, however, shows distinguishable peaks compared with the isolated AzoC6 and αCD powder, indicating the formation of inclusion complexes. In other words, the disappearances of the characteristic endothermic peaks of AzoC6 and αCD, as well as the appearance of the new peaks, provide strong evidence for the AzoC6@2αCD complex formation.

### Macroscopic gel formation via host-guest chemistry

$^1$H NMR spectroscopy is carried out to provide important insights into the inclusion complexation between AzoC6 guests and αCD hosts in the aqueous environment. The chemical structures of AzoC6 and αCD

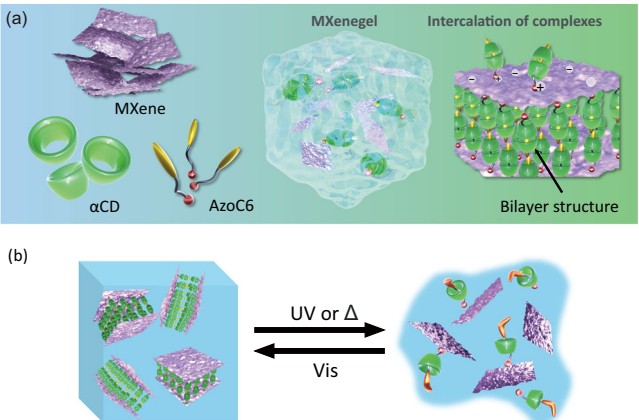

**Fig. 1 | Schematic illustration of the fabrication and microscopic architecture of light-responsive MXenegel. a** The alpha-cyclodextrin (αCD) host and the 1-methyl-3-(5-(4-(phenyldiazenyl)phenoxy)pentyl)-1H-imidazol-3-ium (AzoC6) form a bilayer structure of AzoC6@2αCD complexes between the MXene nanosheets. Meanwhile, the positive head ends of the complexes are electrostatically assembled to the negatively charged MXene surfaces. **b** Light-responsive sol-gel transition behavior of the MXenegel.

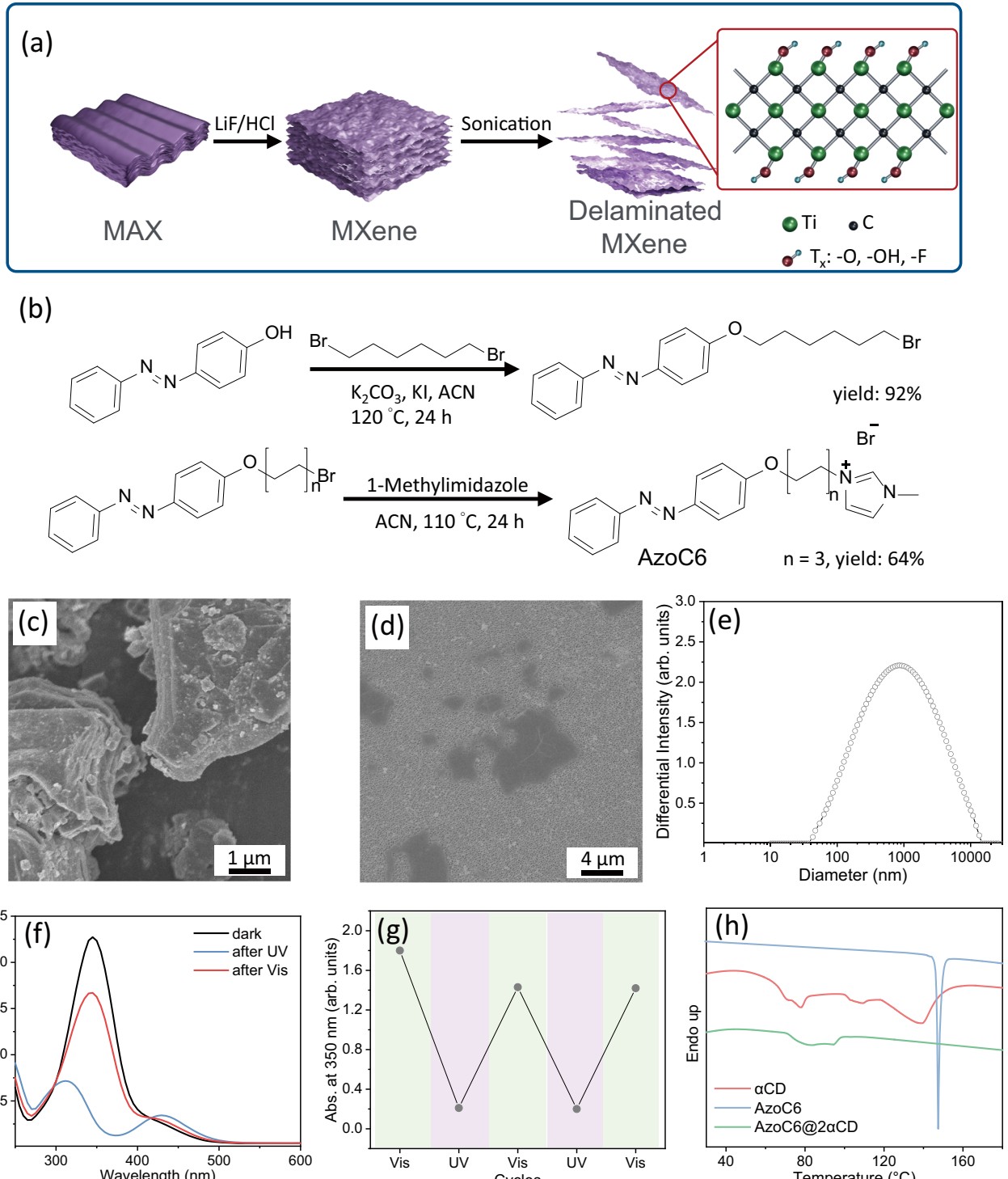

**Fig. 2 | Preparation and characterization of Ti₃C₂Tₓ MXene and 1-methyl-3-(5-(4-(phenyldiazenyl)phenoxy)pentyl)−1H-imidazol-3-ium (AzoC6). a** Fabrication of the delaminated Ti₃C₂Tₓ MXene nanosheets via the MILD method. **b** Synthetic scheme of the AzoC6. SEM images of (**c**) MAX precursor and (**d**) delaminated MXene nanosheets on an anodic aluminum oxide filter. **e** DLS spectrum of the synthesized MXene. The average hydrodynamic diameter is ca. 838 nm. **f** UV−vis spectra of AzoC6@2αCD complex under different light irradiations. **g** Reversible isomerization of AzoC6@2αCD complex under alternative UV and visible light irradiations. The green and purple colors indicate the visible and UV light irradiations, respectively. The UV and visible light irradiation times are 3 and 10 minutes, respectively. **h** DSC heating traces of alpha-cyclodextrin (αCD) powder, AzoC6 powder, and vacuum-dried AzoC6@2αCD complex.

are shown in Fig. 3a, and the corresponding hydrogens of the characteristic peaks marked in Fig. 3b are also denoted. It can be seen from the NMR spectrum of the AzoC6/αCD (1:2 molar ratio) binary system that the peaks are shifted and broadened, indicating both the trans-azobenzene parts and the alkyl chain spacers are included in the αCD

cavities[37–39]. Because of the electromagnetic environmental changes, chemical shifting could happen to both the host and guest molecules. Moreover, the inclusion formation can significantly restrict the guest molecules and decrease their conformational flexibilities, which can be referred to our previous work[34]. Further comprehensive details about

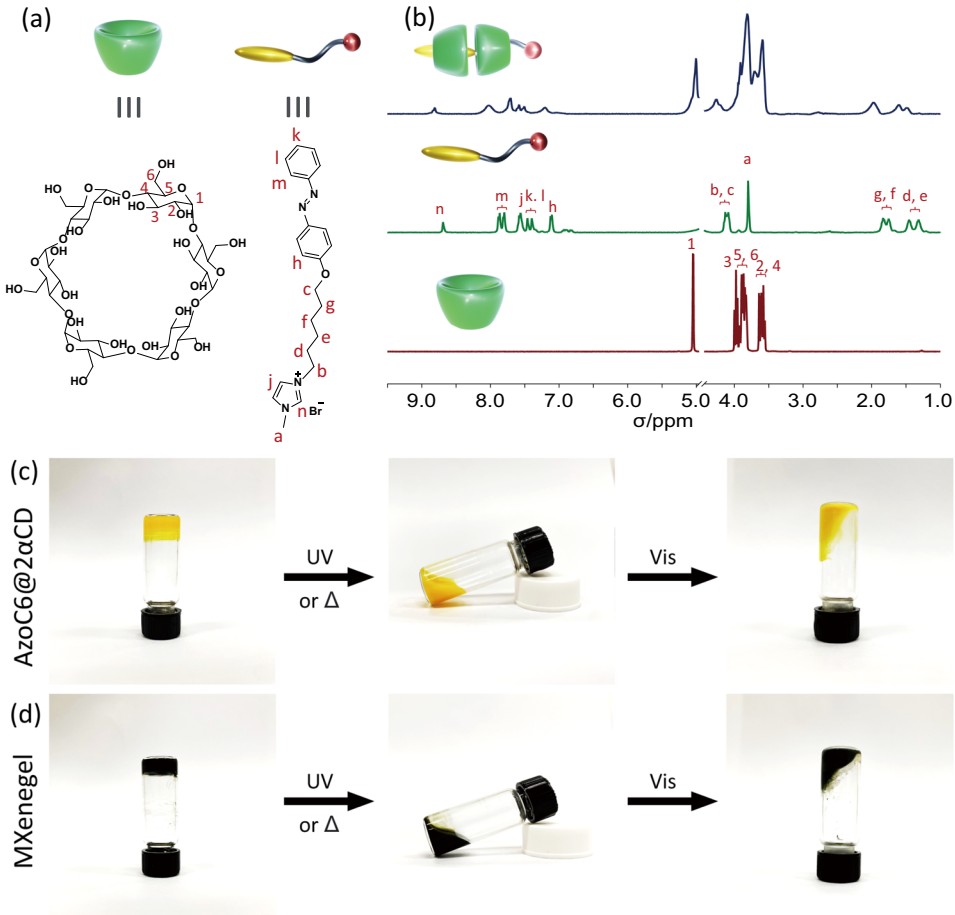

**Fig. 3 | Supramolecular gel formation of AzoC6@2αCD and MXenegel.**
**a** Chemical structures of αCD host and AzoC6 guest molecules. The corresponding hydrogens for NMR spectra are also denoted. **b** ¹H NMR spectra (400 MHz, D₂O, 298 K) of 1-methyl-3-(5-(4-(phenyldiazenyl)phenoxy)pentyl)–1H-imidazol-3-ium (AzoC6), alpha-cyclodextrin (αCD), and AzoC6/αCD complex (1:2 ratio). Light-responsive phase behavior of (**c**) 50 mM AzoC6@2αCD and (**d**) 30 wt% MXenegel under UV and visible light irradiations. The irradiation times of UV and visible light are 10 and 30 min, respectively.

the complexing geometry of αCD and AzoC6 are deduced from the 2D ROESY experiment. The appearances of the correlation peaks are indicative evidence of the specific relationships between adjacent protons. As shown in Supplementary Fig. 8, several cross-peaks, as highlighted with red squares, can be observed, indicating that the specific parts of the different molecules are close to each other in an aqueous environment. In the azobenzene part, multiple correlation signals of benzyl protons with $H_5$ and $H_3$ of αCD are observed, suggesting that the benzene group is embedded into the central cavity of αCD; in the alkyl chain part, cross-peaks of $H_d$, $H_e$, $H_f$, and $H_g$ with inner αCD protons also demonstrate the successful inclusion formation[40].

The macroscopic gelation and light-responsive behavior of Azo-C6@2αCD and MXenegel can be observed by the naked eye. When the AzoC6 and αCD aqueous mixture with the proper 1:2 ratio is incubated in the dark, the solution gradually changes from clear to opaque gel. Supplementary Fig. 9 shows the digital images of the transparent AzoC6 and αCD aqueous solutions. It can be preliminarily concluded that the hydrogel formation is caused by the inclusion of AzoC6 and αCD molecules and the stacking of their inclusion complexes. It is also worth mentioning that this assembly behavior is not affected by the additional doping of MXene nanosheets. As shown in Fig. 3d, when a mixture of MXene, AzoC6, and αCD coexist, an immobile gel is formed. Surprisingly, we observe that the rich functional groups of MXene flakes can strengthen the assembly of the nanostructures. As shown in Supplementary Fig. 10, the liquid-phase DSC analyses of MXenegel show that both glass transition and disassembly temperatures are increased, compared with the undoped gel of AzoC6@2αCD. High-

resolution XPS element scans of the MXene, MXene/AzoC6, and MXenegel are shown in Figs. S12, S13, S14, respectively. For the MXene/AzoC6 sample, even after being washed with DI water several times, the results clearly show that the surfaces of the MXene flakes are attached by plenty of the AzoC6 molecules, as evidenced by the formation of the N1s peak, the C-N peak in the C 1s spectrum, and the C-O peak in the O 1s spectrum; for the MXenegel sample, the strong C-C in C 1s, the C-O peak in O 1s, and the N 1s peak, indicate the successful intercalation of the AzoC6@2αCD inclusions. In addition, it can be confirmed from the high-resolution scan of the Ti 2p spectra that there is no noticeable oxidation of the MXene nanosheets occurring, as the TiO₂ peak remains weak in all of the samples.

Moreover, the effects caused by the illuminations of UV and visible light are remarkable. As shown in Fig. 3c, the Azo@2αCD gel gradually turns into a sol-like structure when under UV light irradiation. Such phase transition mainly originated from the partial decomplexation of the AzoC6@2αCD inclusions. The azobenzene part of the AzoC6 molecule can undergo trans-to-cis photoisomerization under UV light. Because of the unfitted size and increased polarity, the cis-azobenzene parts of the AzoC6 are released from the αCD cavities, while the alkyl chain spacer parts remain included in the αCD hosts. As a result, the gel turns into sol when more Azo@2αCD transforms to Azo@αCD. After irradiating with visible light, the cis-azobenzene parts can isomerize back to trans-azobenzene parts and form inclusions with αCD hosts again. As displayed in Fig. 3d, the light-responsive behavior of the MXenegel is also identical. The MXenegel possesses light-responsiveness even when the MXene concentration is as high as 30 wt

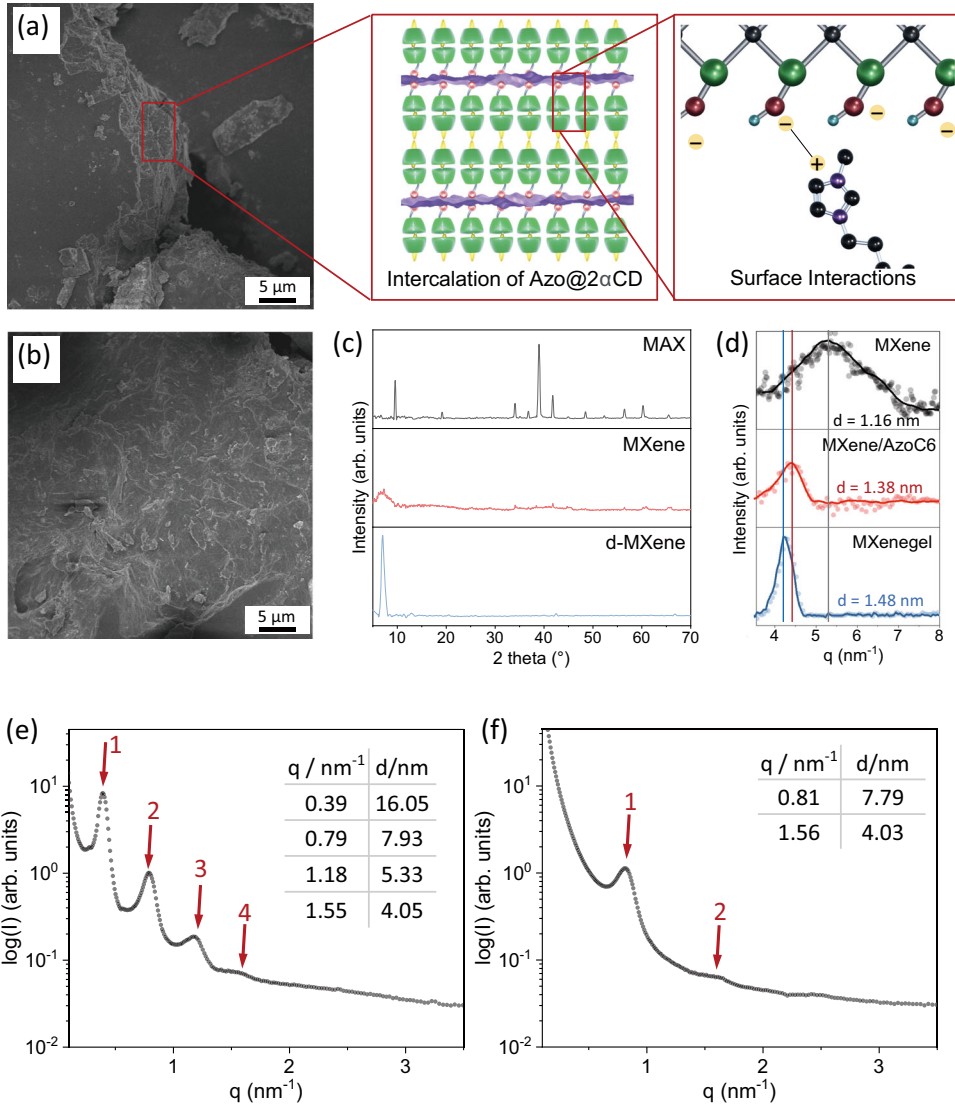

**Fig. 4 | Microstructure of AzoC6@2αCD complexes between the MXene layers.** **a**, **b** SEM images of the vacuum-dried MXenegel (30 wt% of MXene), in which the cross-sectional view of MXenegel layers is highlighted. The positively charged imidazolium groups of the AzoC6@2αCD bilayer are intercalated between the MXene nanosheets and assembled with the negatively charged MXene surfaces. **c** XRD patterns of MAX precursor, MXene powder, and delaminated MXene nanosheets. **d** Magnified WAXS spectra of MXene powder, MXene/AzoC6, and vacuum-dried 30 wt% MXenegel. SAXS spectra of (**e**) AzoC6@2αCD hydrogel and (**f**) 30 wt% MXenegel.

%. In addition, there is no severe precipitation occurring when the MXenegel turns into sol, which is commonly observed in MXene/surfactant aqueous systems[41,42].

**Microstructure of MXenegel revealed by X-ray spectroscopy**

SEM spectroscopy provides a better insight into the surface structure of the vacuum-dried MXenegel. From the SEM images shown in Fig. 4a, b, it can be seen that the MXenegel has a stacking sheet-like structure with AzoC6@2αCD intercalating in between. To give a better insight into the inner microstructure, as well as the MXene flake distribution, vacuum-dried MXenegels with different doping concentrations are also investigated. As shown in Supplementary Fig. 18, the MXene flakes are well-dispersed in the AzoC6@2αCD matrix, forming an interconnecting conductive network throughout the system. The results suggest that the MXene flakes are well-compatible with the AzoC6@2αCD and the microstructure of the MXenegel can be retained at a high doping concentration.

As depicted in Fig. 4a, the driving forces of the formation of the MXenegel can be divided into three parts. First, the AzoC6 and αCD

molecules can form host-guest inclusions with a molar ratio of 1:2 via hydrophobic interaction. Second, the AzoC6@2αCD inclusions aggregate through hydrogen bonding interactions and each layer can self-assemble into a lamella structure in a head-to-head/tail-to-tail order. According to previous research, the self-assembly of the small-molecule/cyclodextrin system mainly depends on the chemical structure of the small molecule and the host/guest ratio. Thus, the ratio between AzoC6 and αCD plays a critical role in this work (Supplementary Fig. 11)[43]. Third, the positively charged imidazolium ends of the AzoC6@2αCD inclusions can attach to the negatively charged surfaces of the MXene nanosheets via Coulombic electrostatic interactions. To study the interactions between AzoC6@2αCD inclusions and MXene in aqueous solution, the zeta potentials of a series of MXene flakes in different concentrations of AzoC6 aquatic solutions are measured. As shown in Supplementary Fig. 12, the surface charges of the MXene flakes increase with the concentration of the AzoC6. The zeta potentials increase from negative values to near zero when the AzoC6 concentration is 0.4 mM, after which a plateau is observed. The result shows that the AzoC6 molecules are tethered to the surfaces of

the MXene flakes, leading to the change of surface charges. ATR-FTIR analyses are also conducted to study the interactions between the AzoC6 molecules and the MXene surfaces. As shown in Supplementary Fig. 19, in the AzoC6 spectrum, the bands at 2933 and 3047, and 1550–1600 cm$^{-1}$ are attributed to C-H symmetric stretching and imidazolium ring stretching, respectively. In the MXene/AzoC6 spectrum, however, the C-H stretching shifts to 2995 and 2916 cm$^{-1}$ and the imidazolium ring stretching shifts to 1599 cm$^{-1}$. Because the absorption of the AzoC6 is significantly larger than that of the MXene flakes, the signals are typically originated from the attached AzoC6. From the zeta potential and ATR-FTIR data, we conclude that the AzoC6 molecules can attach to the surfaces of MXene and the electrostatic interactions between MXene and AzoC6 molecules are strong.

To confirm our hypothesis of the self-assembled microstructure of the MXenegel, we further conduct ex-situ wide-angle X-ray experiments. The diffraction experiments are performed at the X-ray Scattering Beamline 23A1 of the National Synchrotron Radiation Research Center (NSRRC). Figure 4c gives integrated intensity profiles of the MAX precursor, as-prepared MXene powder, and delaminated MXene film. The enlarged regions of the intensity profile, I(q), of the MXene, MXene/AzoC6, and dry MXenegel are displayed in Fig. 4d. The as-prepared MXene exhibits a (002) peak at q = 5.38 nm$^{-1}$ with an interlayer distance of 1.16 nm[44]. The MXene/AzoC6 sample is prepared by adding AzoC6 into the MXene aqueous suspension, followed by vacuum filtration; the MXene/AzoC6 clay is washed with deionized water several times to ensure that the unattached AzoC6 molecules are removed. The (002) peak of the MXene/AzoC6 is at the position of q = 4.54 nm$^{-1}$ with an interlayer distance of 1.38 nm. After the intercalation of the AzoC6 molecules, the (002) peak shifts to a lower angle, suggesting an increase in the interlayer distance for ca. 2.2 Å, which is comparable to the length of a trans-AzoC6 molecule. The (002) peak position of the dry MXenegel is at q = 4.25 nm$^{-1}$ with an interlayer distance of 1.48 nm. The increased interlayer value is ca. 3.2 Å, compared with the pristine MXene. This value is smaller than the double length of AzoC6@2αCD, indicating the collapsing of the bilayer structure during the fast removal of water.

The microstructural details are further evaluated by SAXS analyses. Figure 4e, f shows the integrated intensity profiles of 50 mM Azo@2αCD and 30 wt% MXenegel, respectively. The samples are sealed in polyimide cells and measured at a fixed position in an ambient condition. As for the SAXS profile of Azo@2αCD shown in Fig. 4e, the diffraction maxima are found at q = 0.39, 0.79, 1.18, and 1.55 nm$^{-1}$, corresponding to d-spacings of 16.05, 7.93, 5.33, and 4.05 nm, respectively. The observed q values show a 1:2:3:4 ratio, which is typically found in lamella structures[45,46]. Moreover, the minimum repeated period corresponds to a thickness of 4.05 nm, which also infers that the periodic assemblies are composed of Azo@2αCD bilayers. Interestingly, the lamella structure can be retained when large amounts of MXene are doped into the system, as shown in the diffraction profile of the 30 wt% MXenegel (Fig. 4f). The diffraction maxima obtained in the MXenegel SAXS result are at q = 0.81 and 1.56 nm$^{-1}$, corresponding to d-spacings of 7.79 and 4.03 nm, respectively. As the form factor minimum corresponds to a thickness of 4.03 nm, it is deduced that the inclusion complexes of AzoC6 and αCD molecules are formed and intercalated between the MXene layers. It should be noted that the assembly/disassembly of the AzoC6@2αCD bilayers can only be realized in the presence of water molecules. As shown in Supplementary Fig. 20, the SAXS profile of the vacuum-dried MXenegel exhibits the smallest d-spacing value of 2.07 nm, indicating the collapsing of the bilayer structure. This result is in line with the WAXS analyses that fast water evaporation can induce the destruction of Azo@2αCD bilayers between the MXene. As shown in Supplementary Fig. 21, the maxima disappear after UV irradiation/thermal treatment, indicating the disassembly of the intercalating AzoC6@2αCD bilayers. Because the MXene concentration of the MXenegel is as high

as 30 wt%, we conclude that the photo-and thermal responsive behaviors mainly originate from the AzoC6@2αCD intercalated between the MXene nanosheets.

## MXenegel as writable and reconfigurable conductive inks

We further demonstrate brush printing using the MXenegel (30 wt%) as a conductive ink because of its stability under ambient conditions and fluidity after UV light irradiation or thermal treatment. Unlike conventional conductive ink that requires polymer as an additive to stabilize metallic nanoparticles, MXenegel, even with a high concentration of 30 wt%, no sedimentation is observed over a week (sealed in a centrifuge vial and stored in a refrigerator). The result is because the gel-like nature of MXenegel under low temperatures makes it suitable for long-time storage. After irradiated by UV light or heating in a 50 °C oven, the MXenegel can be used as a conductive ink to print on substrates. Figure 5a shows the NYCU letters written with a Chinese writing brush, and Fig. 5b demonstrates letters of an interconnected NYCU written by 30 wt% of MXenegel ink that can serve as wire to lighten up an LED. Figure 5c displays the in situ current value of the MXenegel wire shown in Fig. 5b under continuous breaking/contacting electromechanical switch. It can be seen from the current value track that the MXenegel wire possesses a fast response time and a stable connecting behavior with no degradation observed. Figure 5d quantifies the MXenegel resistance with the MXene doping content. A pristine 50 mM Azo@2αCD has a conductivity of 6.0 × 10$^{-4}$ S/cm, which is a typical value for ionic conductive hydrogel. The conductivity increases to 1.3 × 10$^{-3}$ S/cm for 30 wt% MXenegel and stabilizes to ca. 1.8 × 10$^{-3}$ S/cm when more MXene is doped. We conclude that both high MXene doping concentration and a homogenous system are necessary for achieving efficient conductance in practical use. Figure 5e displays the representative Nyquist plot of 30 wt% MXenegel under UV and visible light irradiations. The sample shows an increased resistance under visible light irradiation and a decreased value under UV irradiation, caused by the disassembly of Azo@2αCD induces the gel-to-sol of MXenegel, which also increases the mobility. To address the crucial information on the material's durability and long-term applicability, we test a 30 wt% MXenegel with a cyclic electrochemical analysis by irradiating UV and visible lights alternatively (Supplementary Fig. 23). After four full cycles of visible-UV-visible treatment, the conductivity and gel-like behavior of the MXenegel are retained.

In addition to responding to light, temperature responsiveness, which is one of the characteristic natures of hydrogel driven by hydrogen bonding, is also tested. The experiment is conducted by attaching the samples to a thermal control plate and monitored by an electrochemical station as the samples reach desired temperatures (checked by an infrared thermometer). When the temperature is below 30 °C, the 5 wt% MXenegel remains gel-like behavior and exhibits higher impedance values through the scanning frequency range; when the temperature increases to 30–40 °C, the impedance values drop to 200–400 ohm, indicating the disassembly of the Azo@2αCD inclusion aggregates; when the temperature is above 40 °C, the bridging between MXene layers is collapsed and the MXenegel becomes more fluid-like behavior. Consequently, the MXenegel impedances drop to a lower value. The results of the temperature ramping experiment are consistent with the liquid-phase DSC analyses (Supplementary Fig. 10). As shown in Supplementary Fig. 24, we have proposed two kinds of electron/ion transport paths under different physical conditions: at the gel state, the conduction takes place through the electron transport of the MXene flakes and slower ion transport; at the sol state, because of the increased mobility of the molecules, the conduction takes place through the electron transport of the MXene flakes and faster ion transport.

## MXenegel as light-responsive wire in electronic circuit
The unique properties of MXenegel provide an exciting opportunity to be integrated into solid-state electronics as conductive wires or light-

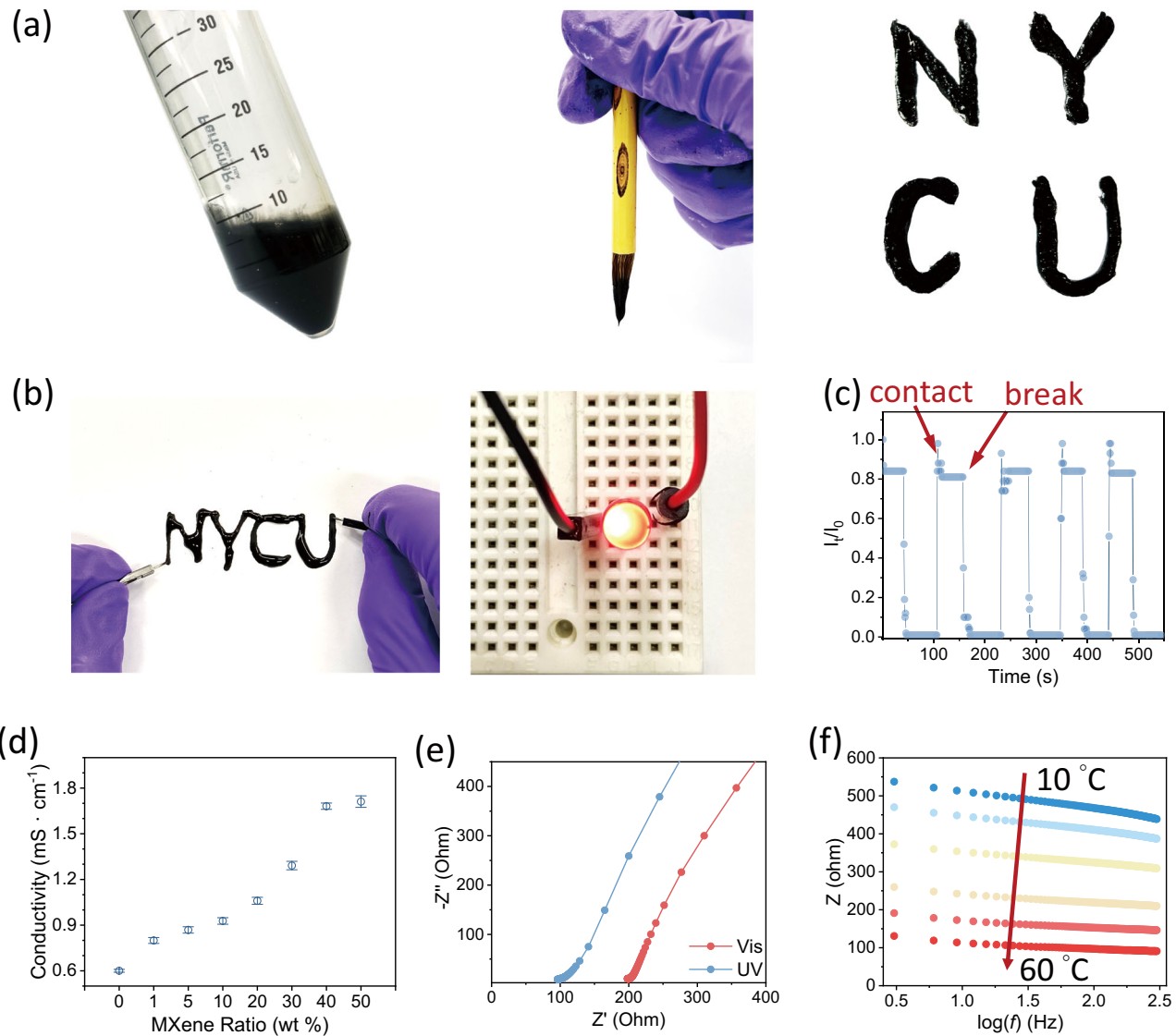

**Fig. 5 | Handwriting of 30 wt% MXenegel inks and electrical performances.**
a Digital images of the MXenegel as conductive inks and the written NYCU. **b** A wire printed by 30 wt% MXenegel can directly lighten up an LED bulb without any amplifier. **c** A long-duration experiment of mechanical cleavage test shows the repeated open and close circuit stability and current change performance of the 30 wt% MXenegel wire. **d** The conductivities of MXenegels with different MXene doping ratios. The error bars are the standard deviations calculated from 10 data points. **e** Nyquist plot of 30 wt% MXenegel after UV and visible light irradiations. **f** Impedance changes of 5 wt% MXenegel over the temperature range of 10–60 °C.

responsive switches, and the electronic circuits can be controlled through a noninvasive approach. Figure 6a shows the schematic illustration of the fabrication and operation of the MXenegel (30 wt%) switch. A transparent tube is first injected with a piece of MXenegel and sealed. The MXenegel inside the tube can stay at the original position upon gently shaking or tilting (Fig. 6b). After UV irradiation, however, the MXenegel becomes fluid-like and flows upon tilting (Fig. 6c, d). When the MXenegel reaches the desired position, it can be solidified again by irradiating with visible light (Fig. 6e). To highlight the efficacy of the MXenegel switch, we create a light-controlled circuit. As shown in the diagram in Fig. 6f, the circuit is integrated with a MXenegel switch, which is connected in series with a pair of parallel LEDs. To connect the MXenegel switch with other electronic objects, both ends of the tube are embedded with a pair of electrodes. When the MXenegel flows to the right side of the tube, the electrodes embedded in the right end of the tube are connected; under this condition, the green LED is on and the red LED is off (Fig. 6g). In the same manner, when the MXenegel flows to the left side of the tube, the electrodes embedded in the left end of the tube are connected; under this condition, the green LED is off and the red LED is on (Fig. 6f). With the incorporation of a MXenegel switch into the circuit, the reversible on/off control of the LEDs through light can be achieved.

The MXenegel embedded switch shown in Fig. 6 demonstrated that not only MXenegel enables response to different wavelengths of light, but also the reconfigurability and the recyclability inherent to MXenegel platforms allows to reprogram or recycle the circuit when it is damaged. During the recycling procedure, the MXenegel can be liquefied through UV light irradiation and separated from other hard electronics or substrates. After the recycling process, the collected MXenegel can be integrated into another circuit. Such abilities provide an exciting opportunity to facilitate electronics for sustainable purposes. The amorphous nature of the MXenegel makes it well-suited for soft electronics, which focuses on the development of flexible and stretchable components and circuits.

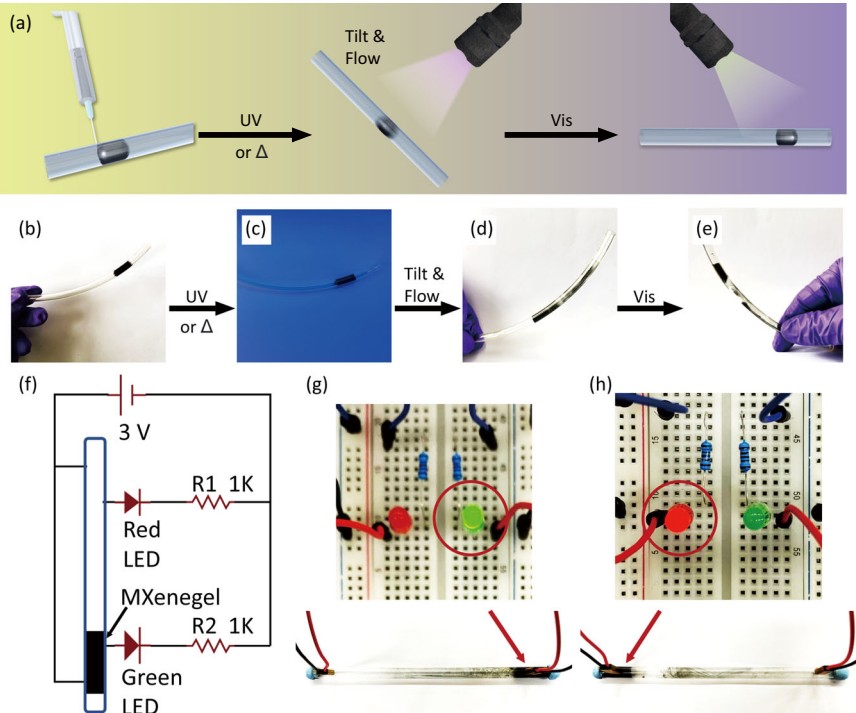

**Fig. 6 | Schematic illustration of the MXenegel as a light-responsive conducting wire. a** The 30 wt% MXenegel is first injected into the transparent tube. After UV light irradiation, the MXenegel becomes sol-like and flows to the other side by tilting the tube. Then, the MXenegel is irradiated with visible light and transforms to gel-like again. **b**–**e** Digital images of the MXenegel wire and the light-responsive phase behavior demonstration. **f** The circuit diagram of MXenegel as an on-demand switch to lighten up LEDs. The MXenegel inside a transparent tube as a switch is also indicated. **g**, **h** After irradiated with UV and visible light, the MXenegel flows to the opposite end embedded with electrodes, by which the on-demand switch control of LEDs can be realized.

Despite the advantages brought by combining Azo/αCD complex and MXene, some potential challenges and limitations still need to be overcome. One major challenge is the environmental tolerance of the MXenegel. Regarding the compatibility of MXenegel in various scenarios, it may require higher temperature resistances and mechanical properties. This issue could be solved by the structural design of the guest molecules that strengthen the interactions with hosts or increase the inclusion ratio to form a stronger stacking array. Another major challenge is the long-term stability that arises from the MXene limitations. MXenes are highly sensitive to environmental conditions, particularly to moisture and oxygen. Because the matrix of the MXenegel is composed of water, the MXenegel may oxidize and lose its conductive function after a long period of storage time. According to previous literature, the Ti atoms in $Ti_3C_2T_x$ MXene may degrade to $Ti^{4+}$ by ca.10 at% after 14 days[47]. In future work, we will focus on developing a nonaqueous-based MXenegel that can encapsulate MXenes with protective coatings and prevent them from being exposed to oxygen.

## Discussion

In summary, we develop a light-modulated MXenegel with reversible phase transition based on photoresponsive host-guest chemistry, which can be used as conducting components and readjusted when needed. Both the azobenzene end and alkyl chain spacer of the AzoC6 molecules can be included in the cavity of αCD host, resulting in a unique nano-building unit of AzoC6@2αCD inclusion. Moreover, the AzoC6@2αCD inclusions are tethered on the surfaces of the MXene nanosheets and self-assembled into a lamella structure between the MXene layers. Detailed NMR, 2D-ROESY, XPS, WAXS, and SAXS are applied to reveal the microstructure of the MXenegel. By irradiating with UV or visible light, the MXenegel can undergo reversible photoresponsive phase transitions, by which the facile reprogrammable or reconfigurable charge transportation paths can be realized. We also

demonstrate this concept by applying the MXenegel into a solid-state circuit to serve as a photo-controllable switch. The concept and strategy proposed in this work aim to build a fundamental groundwork for the creation of MXene-based conductive materials with dynamic functionalities.

## Methods
### Materials

The MAX precursor was obtained from Laizhou Kai Kai Ceramic Materials (>98%). 1-Methylimidazole (99%) and potassium iodide (99%) were purchased from Sigma-Aldrich. 1,6-Dibromohexane (98%) and phenol (99%) were purchased from Acros Organics. Aniline (>99%) was obtained from Alfa Aesar. *alpha*-Cyclodextrin (99%) and lithium fluoride were obtained from Nova Materials. Sodium nitrite (98%) and potassium carbonate (99%) were purchased from Duksan. Hydrochloric acid (36.5 wt%) was purchased from Honeywell. Acetonitrile (99.9%) was obtained from Fisher chemical and purified by a solvent purification system. Deionized water (18.4 MΩ) was obtained from the Milli-Q system. Unless specifically mentioned, all the reagents were used as supplied without further purification.

### Synthesis of MXene nanosheets

$T_3C_2T_x$ MXene was synthesized by the MILD method. Briefly, 29.6 mL of concentrated HCl (37% w/w) was diluted with deionized water to obtain 40 mL of 9 M HCl solution. The HCl solution was transferred to a polypropylene beaker, and 3.2 g of lithium fluoride was added. The mixture was sonicated to dissolve the lithium fluoride. In the ice bath, 2 g of $Ti_3AlC_2$ MAX powder was slowly added to the solution over 15 min. The mixture was stirred vigorously at 20 °C for 24 h. The slurry was collected by centrifugation and washed with deionized water to remove unreacted salts and by-products. The centrifugation-washing process was repeated until the pH was ~6 and the supernatant turned

dark green. The MXene clay was dried and collected from the bottom of the centrifuge tube. The crude MXene powder was stored in an inert and dried atmosphere.

## Preparation of MXenegel

The AzoC6@2αCD and MXenegel samples were prepared by combining MXene nanosheets with desired amounts of AzoC6 and αCD to give a constant AzoC6/αCD molar ratio of 1:2. Take preparing 30 wt% of MXenegel as an example. First, 110.8 mg AzoC6 (0.25 mmol) powder was added to 2 mL deionized water and stirred to dissolve. Subsequently, 2.36 g MXene and 486.5 mg αCD (0.5 mmol) were added into another glass vial with 3 mL deionized water. Then, the MXene/αCD mixture was sonicated for 20 min to dissolve the αCD and to separate the MXene layers. The gelation and inclusion formation were carried out by slowly dropping the AzoC6 solution into the MXene/αCD mixture and incubating for 2 h at 20 °C in a dark room. The concentrations of MXene and AzoC6 of the MXenegel are expressed as weight percent and molarity, respectively.

## Characterizations

The surface morphologies were characterized by a scanning electron microscope (SEM, JEOL JSM-7401F) at an acceleration voltage of 5 kV. The samples were coated with a thin layer of platinum (~4 nm) before the measurements. UV−vis absorption spectra were measured from 250 to 850 nm using a Hitachi U-4100 spectrometer. X-ray photoelectron (XPS) spectra were recorded on a Thermo VG Scanning Auger XPS/AES Microlab 350 (VG Scientific) using Al Kα radiation as the excitation source. For MXene and MXene/AzoC6 XPS spectra, the samples were prepared in the aqueous state, filtered by a 200 nm filter paper, and washed with DI water several times to remove any possible residue and the unattached AzoC6 molecules; for the MXenegel spectra, the samples were prepared by the vacuum-dried method. $^1$H NMR spectra were collected on a ECZ500R/S1 NMR spectrometer (JOEL). 2D ROESY experiments were acquired with a mixing time of 600 ms. ESI(+)(-)MS experiments were carried out using a Impact HD Q-TOF mass spectrometer (Bruker) equipped with an electrospray ionization (ESI) source operating in positive ion mode. The parameters of ESI(+) included 4.5 kV for ion spray voltage, 200 °C for capillary temperature, and 6 L/min for sheath gas flow rate. The SAXS data were collected at the X-ray Scattering Beamline 23A1 of National Synchrotron Radiation Research Center (NSRRC) located at Hsinchu, Taiwan. The radiation wavelength was 1.03321 Å. The powder-like samples were characterized by a Bruker APEX II diffractometer equipped with an area detector (CCD). The radiation wavelength was 1.5406 Å. The integrated scattering intensity profiles were plotted versus the scattering vector (q), where q was defined as $q = 4\pi\sin(\theta/2)/\lambda$ and θ and λ represent the scattering angle and the wavelength, respectively. The hydrodynamic radius of the MXene nanosheets were determined using an Otsuka ELSZ-2000S particle size distribution analyzer with a DLS mode. A differential scanning calorimeter (DSC Q200, TA Instruments) equipped with a refrigerated cooling system (RCS90) was conducted for thermal analyses. Rheological measurements were collected by a Discovery HR-2 (TA Instruments) equipped with a Peltier plate temperature-controlling system. An 8 mm stainless steel plate geometry was used in our experiments. The axial force for sample compression was set at 1 N and the frequency region was from 0.01 to 100 Hz. The operating temperatures were kept at the desired values with the accuracy controlled to ±0.2 °C. The optimized structures are obtained by DFT calculations using Gauss view and Gaussian 09 package in B3LYP/6-31 G(d, p) basis set at 298.15 K.

## Electrochemical measurements

The electrochemical measurements were carried out by an LCR meter (LCR-6000, GWInstek). A frequency range of 10 Hz–300 kHz was used, and the fitting analyses were performed using ZView2 software. Each

data point was calculated by at least 10 measurements. The UV (365–375 nm, 300–700 mW) and visible (5600–6300 K, 700 mW) light sources used in this work were obtained from Vitastar. To measure the conductivities under different light irradiations, the samples were sandwiched between two ITO glasses and separated by a piece of Teflon spacer (~1 mm). The temperature ramping experiments were conducted by placing the samples on a thermal control plate. To ensure proper thermal conduction, a piece of thermal tape was applied to the surfaces of the samples in contact with the plate. For the MXenegel wire demonstration, ~1 mL of MXenegel was injected into a transparent tube, on which two pairs of electrodes were embedded at the opposite ends of the tube. After both sides of the tube were sealed, the experiments were conducted by irradiating with different light sources.

## Data availability

The DLS, SAXS, WAXS, UV-Vis, Rheology, FTIR, and DLS data generated in this study have been deposited in the Figshare database [https://doi.org/10.6084/m9.figshare.24803037]. The processed XPS, HR-ESI, and NMR data are provided in the Supplmentary Information. The data that support the findings of this study are also available from the corresponding author upon request.

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

## Acknowledgements

This work is supported by the 2030 Cross-Generation Young Scholars Program of the National Science and Technology Council, Taiwan (NSTC) under Grant No. MOST 111-2628E-A49-021 and NSTC 112-2628-E-A49-012, and the Center for Emergent Functional Matter Science of National Yang Ming Chiao Tung University from the Featured Areas Research Center Program within the framework of the Higher Education Sprout Project by the Ministry of Education (MOE) in Taiwan. We thank Dr. You-Song Cheng at the NYCU Instrumentation Resource Center for the assistance in NMR experiments.

## Author contributions

Y.L.L. conceived the project. S.Z. and C.C.C. synthesized the materials and prepared the Azo@2αCD samples. Y.L.L., S.Z., and L.R.L. performed the characterizations and electrochemical measurements. J.T.C. acquired the funding and supervised the project. Y.L.L. and J.T.C. wrote the manuscript with contributions from all co-authors. All authors discussed the results and comments on the manuscript.

## Competing interests

The authors declare no competing interests.

## Additional information

**Peer review information** : *Nature Communications* thanks Kuniharu Ijiro, Sungjun Park and the other anonymous reviewer(s) for their contribution to the peer review of this work. A peer review file is available.

