## [Peer Review File · Nature Communications]

Light-Responsive MXene/gel via Interfacial Host-Guest Supramolecular BridgingREVIEWER COMMENTS

Reviewer #1 (Remarks to the Author)

Please see the attachment.

Reviewer #2 (Remarks to the Author):

Reviewer Comment:

Overall, this paper presents the development of a light-modulated MXene gel with reversible phase transition using photoresponsive host-guest chemistry. The authors demonstrate the inclusion of both the azobenzene end and alkyl chain spacer of AzoC6 molecules in the cavity of α CD host, resulting in a unique nano-building unit. These units are then tethered on the surfaces of MXene nanosheets and self-assembled into a lamella structure. The microstructure of the MXene gel is characterized using various techniques such as NMR, 2D-ROESY, XPS, WAXS, and SAXS. The MXene gel exhibits reversible photo-responsive phase transitions upon irradiation with UV or visible light, allowing for facile reprogrammable or reconfigurable charge transportation paths. The authors demonstrate the application of the MXene gel as a photo-controllable switch in a solid-state circuit.

Overall, this work introduces a novel concept and strategy for the creation of MXene-based conductive materials with dynamic functionalities. However, the paper lacks clarity in certain areas, and the authors could provide more detailed explanations of the experimental methods and results. Additionally, the significance and potential impact of the findings are not sufficiently emphasized. Furthermore, the experimental results should be presented with more statistical analysis and replication to strengthen the reliability of the findings. Overall, while the concept is interesting, the paper requires significant revisions to improve its clarity, significance, and robustness.

#Comment 1:

The author demonstrates the reversible isomerization of the AzoC6@2 α CD complex through the use of UV and visible light irradiation for 3 and 10 minutes, respectively (Fig. 2g). However, the manuscript lacks a detailed explanation of the differences in intensity between UV and visible light as observed in the UV-vis spectrum (Fig. 2f) and why the visible light condition requires a longer irradiation time. Therefore, it is important to provide a more comprehensive description of these aspects in the manuscript. Specifically, it should explain the absorption characteristics of the AzoC6@2 α CD complex under both UV and visible light conditions. The manuscript should clarify whether the observed differences in intensity are due to variations in the absorption cross-sections or the molar absorptivities of the complex for UV and visible light wavelengths. Additionally, the manuscript should discuss any potential shifts in the absorption peaks or changes in the electronic transitions that occur as a result of the isomerization process induced by UV and visible light. Also, please unify units for light condition for UV (mW \diamond mW/cm²) and visible light (lm \diamond mW/cm²) in method section.

#Comment 2:

In the case of the MXene gel depicted in Figure 3, it is important to investigate how the conductivity of the material can vary depending on the concentration with phase condition (solid and gel). The manuscript should provide a more detailed explanation of this relationship to elucidate the conductivity behavior of the MXene gel with phase status (fig 5d). The manuscript should discuss the experimental methods used to measure the conductivity at different concentrations of the MXene gel. It should highlight the specific concentrations

tested and how the conductivity values were obtained. Additionally, the manuscript should present the results graphically, illustrating the relationship between concentration and conductivity.

#Comment 3:

In the case of the MXene gel depicted in Figure 3, it is essential for the author to clarify the number of times the phase transition between solidification and liquefaction can be repeated. This information is crucial in assessing the material's durability and long-term applicability. The manuscript should include a comprehensive explanation of the reversibility of the phase transition in the MXene gel. It should specify the number of cycles or repetitions in which the solidification and liquefaction process can be performed without significant degradation or loss of functionality. This could be determined through experimental measurements or by providing an estimation based on the material's stability and previous observations.

#Comment 4:

The author needs to provide a description of how the mechanical properties of the MXene gel can differ under UV and visible light irradiation, specifically highlighting how the material is strengthened after the phase transformation. This information is crucial for understanding the impact of light on the mechanical behavior of the MXene gel. The manuscript should discuss the experimental methods used to evaluate the mechanical properties of the MXene gel under both UV and visible light irradiation conditions. This may involve techniques such as tensile testing, nanoindentation, or other relevant mechanical characterization methods. The manuscript should also provide details regarding the specific parameters employed during these experiments.

#Comment 5:

In the context of conductivity (as depicted in Figures 5 and 6), it is crucial to address the compatibility of thick film with electronic circuit applications. The manuscript should provide a refined statement on this matter. The manuscript should discuss the implications of thick film conductivity for electronic circuit applications. It should highlight the potential challenges or limitations associated with using thick films in practical electronic circuitry. This could involve considerations such as increased resistance, difficulties in achieving precise patterning, or limitations in miniaturization due to the thickness of the film. Also describe key application of MXene gel for the use of electronic applications.

Reviewer #3 (Remarks to the Author):

This paper describes the intercalation of azobenzene-containing supramolecular inclusion complexes into MXene nanosheets to form photo-responsive conducting gels. AzoC6@2 α CD inclusion complexes bind to the surface of MXene nanosheets and self-assemble into lamellar structures between MXene layers. Upon irradiation with UV or visible light, MXene gel undergoes a reversible photo-responsive phase transition. The application of MXene gel to solid-state circuits and to optical controllable switches is very interesting. However, MXene gel's explanation of reversible phase transitions is insufficient, and the reviewer would claim major revision. Specific points are shown below:

There are questions about the mechanism of the reversible photo-responsive phase transition in MXene gel. As shown in Figure 3c, the Azo@2 α CD gel gradually changes to a sol-like structure when irradiated with UV light. The photo-responsive behavior of MXene gel

is also identical, as shown in Figure 3d. In the case of Figure 3d, this may be due to the properties of free AzoC6@2 α CD, which is not intercalated into the MXene nanosheet. For this to be the case, the change in thickness of the MXene nanosheet before and after UV irradiation must be clearly shown.

In terms of supramolecular chemistry, it should explain why the bilayer structure of the AzoC6@2 α CD complex is broken when Azo@2 α CD is changed to Azo@ α CD by UV irradiation.

General comments:

With the development of technology and the emphasis on ecological issues, new demands have been placed on electronic components in terms of their environmental sensitivity and recyclability. In this work, Lin et al. design a light-modulated MXene gel, that undergoes the gel-sol transition under UV light irradiation or heat treatment, and demonstrate unique applications in a solid-state circuit as a photo-controllable switch. Based on the special structure and the host-guest chemistry, the AzoC6 molecules can be incorporated into the cavity of the α CD host to form a unique nano-building unit of AzoC6@2 α CD inclusion; meanwhile, this inclusion can self-assemble into a lamellar structure in between the MXene layers, just like the lipid bilayer in the cell membrane.

This is a well written, and elegant manuscript, to describe the interfacial bridging process of the supramolecular in MXene interlayers and the reversible photo-modulated phase behavior of the obtained MXene gel. The authors have provided solid experimental data, and demonstrated good applicability of this design. The microscopic architecture of this gel is very special and rarely reported; the application is also novel and attractive, which will be of great interest to researchers in the MXene-related electronic fields. Therefore, the manuscript is recommended for publication at the high professional level of *Nature Communications* after addressing some minor issues. The detailed comments are given below:

1. It is recommended that the authors mark the glass transition or disassembly temperatures in Figure S6. At what temperature does this gel-sol transition occur?
2. It will be helpful if a high-resolution SEM or TEM image of the MXene gel can be provided.
3. The light-responsive phase behavior of AzoC6@2 α CD and the MXene gel is very interesting. The AzoC6@2 α CD disintegrates under UV light irradiation, and what about the interactions between AzoC6/ α CD and MXene? Any relevant NMR or FTIR spectra will be helpful.
4. How long can this MXene gel be stored with its nature preserved? And what are the rheological properties (e.g., viscosity or modulus) of this gel?
5. The authors added MXene to the AzoC6/ α CD solution at a concentration of 30 wt.%. Does the dose of MXene influence the microscopic architecture of MXene gel? Some related discussions are needed.
6. As the authors mentioned the recyclability of electronic components, it will be better with some related experiments or demonstration as a proof of this concept.
7. Authors should give the full names for the abbreviations, when they appear for the first time. It will be better that the figures can be more standardized and nicer.

Point-by-point response to the reviewers' comments

REVIEWER COMMENTS

Reviewer #1:

General comments:

With the development of technology and the emphasis on ecological issues, new demands have been placed on electronic components in terms of their environmental sensitivity and recyclability. In this work, Lin et al. design a light-modulated MXene gel, that undergoes the gel-sol transition under UV light irradiation or heat treatment, and demonstrate unique applications in a solid-state circuit as a photo-controllable switch. Based on the special structure and the host-guest chemistry, the AzoC6 molecules can be incorporated into the cavity of the α CD host to form a unique nano-building unit of AzoC6@2 α CD inclusion; meanwhile, this inclusion can self-assemble into a lamellar structure in between the MXene layers, just like the lipid bilayer in the cell membrane.

> We thank the reviewer for summarizing the main concept of this work.

This is a well-written, and elegant manuscript, to describe the interfacial bridging process of the supramolecular in MXene interlayers and the reversible photo-modulated phase behavior of the obtained MXene gel. The authors have provided solid experimental data and demonstrated good applicability of this design. The microscopic architecture of this gel is very special and rarely reported; the application is also novel and attractive, which will be of great interest to researchers in the MXene-related electronic fields. Therefore, the manuscript is recommended for publication at the high professional level of Nature Communications after addressing some minor issues. The detailed comments are given below:

> We thank the reviewer's positive comments on the quality of this work. We believe that this work provides an in-depth understanding of photoresponsive conductive composites. Also, this work aims to raise a fresh perspective on bridging MXene flakes and supramolecular complexes, as well as their potential applications in sustainable electronics.

1. It is recommended that the authors mark the glass transition or disassembly temperatures in Figure S6. At what temperature does this gel-sol transition occur?

> As suggested by the reviewer, we have marked the glass transition and disassembly temperatures. The glass transition temperatures of the Azo@2 α CD and the Azo@2 α CD-30 wt% are 28 and 40 °C, respectively. Also, the gel-to-sol transition temperatures of the Azo@2 α CD and the Azo@2 α CD-30 wt% are 43 and 51 °C, respectively. We have marked both the glass transition and gel-to-sol transition temperatures in the DSC spectra.

In the revised Supporting Information, we have modified Figure S7 and mentioned the glass transition and disassembly temperatures in the caption.

Figure S1. DSC heating traces of the AzoC6@2 α CD hydrogel and the 30 wt% MXene gel. The glass transition and disassembly temperatures are indicated.

2. It will be helpful if a high-resolution SEM or TEM image of the MXene gel can be provided.

> We agree with the comment from the reviewer that a high-resolution image of MXene gel can provide a

better insight into the material. To reveal the inner microstructure, as well as the MXene flake distribution, SEM is applied to investigate the vacuum-dried MXenegel with different doping concentrations. As shown in the newly obtained SEM images, the MXene flakes are well-dispersed and interconnected in the matrix, forming a stacking sheet-like structure with AzoC6@2 α CD inclusions inserted between the layers.

In the revised Supporting Information, we have added the following SEM images as Figure S15 to provide the inner structures of the vacuum-dried MXenegels with different MXene concentrations.

Figure S15. SEM images of vacuum-dried (a) 0.25, (b) 1.25, (c) 5, and (d) 30 wt% MXenegels.

In the revised manuscript, we have also revised the following sentences to discuss more on the microstructure of the MXenegel based on the newly obtained SEM images.

SEM spectroscopy provides a better insight into the surface structure of the vacuum-dried MXenegel. From the SEM images shown in Figure 4a,b, it can be seen that the MXenegel has a stacking sheet-like structure with AzoC6@2 α CD intercalating in between. To give a better insight into the inner microstructure, as well as the MXene flake distribution, vacuum-dried MXenegels with different doping concentrations are also investigated. As shown in Figure S15, the MXene flakes are well-dispersed in the AzoC6@2 α CD matrix, forming an interconnecting conductive network throughout the system. The results

suggest that the MXene flakes are well-compatible with the AzoC6@2 α CD and the microstructure of the MXene gel can be retained at a high doping concentration.

3. The light-responsive phase behavior of AzoC6@2 α CD and the MXene gel is very interesting. The AzoC6@2 α CD disintegrates under UV light irradiation, and what about the interactions between AzoC6/ α CD and MXene? Any relevant NMR or FTIR spectra will be helpful.

> To study the interactions between AzoC6@2 α CD inclusions and MXene in aqueous solution, the zeta potentials of a series of MXene flakes in different concentrations of AzoC6 aqueous solution are measured. Based on the obtained data, in general, the surface charges of the MXene flakes increase with the concentration of the AzoC6. The zeta potential increases from negative values to near zero when the AzoC6 concentration is 0.4 mM, after which a plateau is observed. The result shows that the AzoC6 molecules are tethered to the surfaces of the MXene flakes, leading to the change of surface charges. Also, the rapid increase of the values indicates that the electrostatic interactions between MXene and AzoC6 molecules are strong.

In the revised Supporting Information, we have added the following zeta potential data as Figure S9.

Figure S9. Zeta potential of MXene (0.5 mg/mL) at different concentrations of AzoC6 aqueous solution.

We also conduct the ATR-FTIR analyses to study the interactions between the AzoC6 molecules and the MXene surfaces. In the AzoC6 spectrum, the bands at 2933 and 3047, and 1550-1600 cm^{-1} are attributed to C-H symmetric stretching and imidazolium ring stretching, respectively. In the MXene/AzoC6 spectrum, however, the C-H stretching shifts to 2995 and 2916 cm^{-1} , and the imidazolium ring stretching shifts to 1599 cm^{-1} . Because the absorption of the AzoC6 is significantly larger than that of the MXene flakes, the signals are typically originated from the attached AzoC6. We conclude that the AzoC6 molecules can attach to the surface of MXene and the electrostatic interactions between MXene and AzoC6 are strong.

In the revised Supporting Information, we have added the following ATR-FTIR data as Figure S16.

Figure S16. ATR-FTIR spectra of the AzoC6 powder and MXene/AzoC6 thin film: (a) C-H stretching and (b) imidazole ring stretching modes. The MXene/AzoC6 sample is rinsed with DI water several times to remove the unattached AzoC6 molecules.

In the revised manuscript, we have added the following sentences to mention the results of zeta potential and ATR-FTIR.

To study the interactions between AzoC6@2 α CD inclusions and MXene in aqueous solution, the zeta potentials of a series of MXene flakes in different concentrations of AzoC6 aquatic solutions are measured. As shown in Figure S9, the surface charges of the MXene flakes increase with the concentration of the AzoC6. The zeta potentials increase from negative values to near zero when the AzoC6 concentration is 0.4 mM, after which a plateau is observed. The result shows that the AzoC6 molecules are tethered to the surfaces of the MXene flakes, leading to the change of surface charges. ATR-FTIR analyses are also conducted to study the interactions between the AzoC6 molecules and the MXene surfaces. As shown in Figure S16, in the AzoC6 spectrum, the bands at 2933 and 3047, and 1550-1600 cm^{-1} are attributed to C-H symmetric stretching and imidazolium ring stretching, respectively. In the MXene/AzoC6 spectrum, however, the C-H stretching shifts to 2995 and 2916 cm^{-1} and the imidazolium ring stretching shifts to 1599 cm^{-1} . Because the absorption of the AzoC6 is significantly larger than that of the MXene flakes, the signals are typically originated from the attached AzoC6. From the zeta potential and ATR-FTIR data, we conclude that the AzoC6 molecules can attach to the surfaces of MXene and the electrostatic interactions between MXene and AzoC6 molecules are strong.

4. How long can this MXenegel be stored with its nature preserved? And what are the rheological properties (e.g., viscosity or modulus) of this gel?

> To address the crucial information on the material's durability and long-term applicability, first, we test a 30 wt% MXenegel with a cyclic electrochemical analysis by irradiating UV and visible lights alternatively. After four full cycles of visible-UV-visible treatment, the conductivity and gel-like behavior of the MXenegel are retained. The long-term stability, however, is a challenging issue in terms of the oxidating nature of the $\text{Ti}_3\text{C}_2\text{T}_x$ MXene in an aqueous system. According to previous literature, the Ti atoms in $\text{Ti}_3\text{C}_2\text{T}_x$ MXene may degrade to Ti^{4+} by ca.10 at% after a 14-day storage period in water. [*Adv. Mater. Interfaces* **2022**, 9, 2200480] Because the matrix of the MXenegel is also composed of water, it is possible that the MXenegel can lose its conductive function after a long storage time.

In the revised Supporting information, we have added the following chart with digital images as Figure S20.

Figure S20. Conductivity analysis of the 30 wt% MXene gel under alternating UV and visible irradiations. The images of the MXene gel at the 0th, 2nd, and 4th cycles are shown.

In the revised manuscript, we have modified the following sentences to mention the cyclic test shown in Figure S20.

To address the crucial information on the material's durability and long-term applicability, we test a 30 wt% MXene gel with a cyclic electrochemical analysis by irradiating UV and visible lights alternatively (Figure S20). After four full cycles of visible-UV-visible treatment, the conductivity and gel-like behavior of the MXene gel are retained.

In the revised manuscript, we have added the following sentences to discuss the challenges of long-term stability and the possible solution of the MXene gel.

Another major challenge is the long-term stability that arises from the MXene limitations. MXenes are highly sensitive to environmental conditions, particularly to moisture and oxygen. Because the matrix of the MXene gel is composed of water, the MXene gel may oxidize and lose its conductive function after a long period of storage time. According to previous literature, the Ti atoms in Ti₃C₂T_x MXene may degrade to Ti⁴⁺ by ca.10 at% after 14 days. [Adv. Mater. Interfaces 2022, 9, 2200480] In future work, we will focus on developing a nonaqueous-based MXene gel that can encapsulate MXenes with protective coatings and prevent them from being exposed to oxygen.

Also, in the revised manuscript, we have added the following reference as ref. 47.

47. Zhao, X. et al. *The Role of Antioxidant Structure in Mitigating Oxidation in Ti3C2Tx and Ti2CTx MXenes. Advanced Materials Interfaces* **9**, 2200480 (2022).

As for the rheological properties, in the revising process, we utilize a rheometer to investigate the MXenegel under different temperatures in the desired frequency range, as shown in the following figure. Because of the instrument limit, we have conducted the temperature ramping experiments. We use an 8 mm stainless steel plate geometry equipped with a water-circulating temperature-controlled system. From the obtained results, the G' and G'' of the 30 wt% MXenegel have dropped by nearly 3 orders of magnitude when the temperature increases from 25 to 50 °C. Similar viscosity change can also be observed in the operating shearing range. This is because the self-assembled nanostructure of the AzoC6@2 α CD is mainly driven by the hydrogen bonds between the neighboring α CDs and the hydrophilic heads of the AzoC6 molecules. When the temperature increases above 50 °C, the nanostructure disassembles. The results obtained from the rheological measurements are in good agreement with the DSC results, in which the disassembly temperature at 51 °C is observed.

In the revised Supporting Information, we have added the following rheological data as Figure S10.

Figure S10. (a) Frequency sweep and (b) viscosity shear rate data of the 30 wt% MXenegel at different temperatures.

In the revised Supporting Information, we have added the following sentences to discuss more the rheological properties of the MXene gel.

The temperature ramping experiments are conducted in an 8 mm stainless steel plate geometry equipped with a water-circulating temperature-controlled system. The results shown in Figure S10 exhibit the changes of the storage modulus G' and loss modulus G'' and viscosity as a function of frequency oscillating frequency and shear rate, respectively. The G' and G'' of the 30 wt% MXene gel have dropped by nearly 3 orders of magnitude when the temperature increases from 25 to 50 °C. Similar viscosity change can also be observed in the operating shearing range. This is because the self-assembled nanostructure of the AzoC6@ α CD is mainly driven by the hydrogen bonds between the neighboring α CDs and the hydrophilic heads of the AzoC6 molecules. When the temperature increases above 50 °C, the nanostructure disassembles. The results obtained from the rheological measurements are in good agreement with the DSC results, in which the disassembly temperature at 51 °C is observed.

Also, in the revised Supporting Information, we have provided details regarding the specific parameters employed during these experiments.

Rheological measurements were collected by a Discovery HR-2 (TA Instruments) equipped with a Peltier plate temperature-controlling system. An 8 mm stainless steel plate geometry was used in our experiments. The axial force for sample compression was set at 1 N and the frequency region was from 0.01 to 100 Hz. The operating temperatures were kept at the desired values with the accuracy controlled to ± 0.2 °C.

5. The authors added MXene to the AzoC6/ α CD solution at a concentration of 30 wt.%. Does the dose of MXene influence the microscopic architecture of MXene gel? Some related discussions are needed.

> To understand whether the dose of MXene influences the microscopic architecture of MXene gel, high-resolution images of MXene gel can provide a better insight into the material. To reveal the inner microstructure, as well as the MXene flake distribution, SEM is applied to investigate the vacuum-dried and freeze-dried MXene gel with MXene doping concentration from 0.25 to 30 wt%. As shown in the newly obtained SEM images, the MXene flakes are well-dispersed and interconnected in the matrix,

forming a stacking sheet-like structure with AzoC6@2 α CD inclusions inserted between the layers. The results show that the MXene flakes are well-compatible with the AzoC6@2 α CD and the microstructure of the MXene gel can be retained at a high doping concentration.

In the revised Supporting Information, we have added the following SEM images as Figure S15 to provide the inner structures of the vacuum-dried MXene gels with different MXene concentrations.

Figure S15. SEM images of vacuum-dried (a) 0.25, (b) 1.25, (c) 5, and (d) 30 wt% MXene gels.

In the revised manuscript, we have also revised the following sentences to discuss more on the microstructure of the MXene gel based on the newly obtained SEM images.

SEM spectroscopy provides a better insight into the surface structure of the vacuum-dried MXene gel. From the SEM images shown in Figure 4a,b, it can be seen that the MXene gel has a stacking sheet-like structure with AzoC6@2 α CD intercalating in between. To give a better insight into the inner microstructure, as well as the MXene flake distribution, vacuum-dried MXene gels with different doping concentrations are also investigated. As shown in Figure S15, the MXene flakes are well-dispersed in the AzoC6@2 α CD matrix, forming an interconnecting conductive network throughout the system. The results suggest that the MXene flakes are well-compatible with the AzoC6@2 α CD and the microstructure of the

MXenegel can be retained at a high doping concentration.

6. As the authors mentioned the recyclability of electronic components, it will be better with some related experiments or demonstration as a proof of this concept.

> The MXenegel embedded switch shown in Figure 6 demonstrates that not only MXenegel enables response to different wavelengths of light, but also the reconfigurability and recyclability inherent to MXenegel platforms allows to reprogram or recycle the circuit when it is damaged. During the recycling procedure, the MXenegel can be liquefied through UV light irradiation and simply separated from other hard electronics or substrates. Instead of fabricating a new one, the collected MXenegel is able to integrate into another circuit or be stored. Such abilities provide a new opportunity to facilitate electronics for sustainable purposes.

In the revised manuscript, we have added the following sentences to emphasize the potential of MXenegel in recyclable and sustainable electronics.

The MXenegel embedded switch shown in Figure 6 demonstrated that not only MXenegel enables response to different wavelengths of light, but also the reconfigurability and the recyclability inherent to MXenegel platforms allows to reprogram or recycle the circuit when it is damaged. During the recycling procedure, the MXenegel can be liquefied through UV light irradiation and separated from other hard electronics or substrates. After the recycling process, the collected MXenegel can be integrated into another circuit. Such abilities provide a new opportunity to facilitate electronics for sustainable purposes. The amorphous nature of the MXenegel makes it well-suited for soft electronics, which focuses on the development of flexible and stretchable components and circuits.

7. Authors should give the full names for the abbreviations, when they appear for the first time. It will be better that the figures can be more standardized and nicer.

> We appreciate the reviewer for pointing out that the abbreviations and the figures should be more

standardized and informative with the names. In the revising process, we have made a number of corrections and modifications which are listed below.

1. In the Introduction section, "... undergo a liquefaction process within seconds under NIR laser irradiation..." has been modified to "... undergo a liquefaction process within seconds under near-infrared (NIR) laser irradiation..."
2. In the Introduction section, "... azobenzene moieties are first included in the cavities of α CD hosts ..." has been modified to "... azobenzene moieties are first included in the cavities of *alpha*-cyclodextrin (α CD) hosts ...".
3. In the Introduction section, "... is prepared by weighing a desired amount of AzoC6, α CD, and MXene into the ..." has been modified to "... is prepared by weighing a desired amount of azobenzene derivative (AzoC6), α CD, and MXene into the ...".
4. In the caption of Figure 1, "...The α CD host and AzoC6 guest molecules form a bilayer structure..." has been modified to "...The α CD host and azobenzene derivatives (AzoC6) form a bilayer structure..."
5. In Figure 2b, we have added the abbreviations under the specific chemical structures.

Modified Figure 2

Reviewer #2 (Remarks to the Author):

Reviewer Comment:

Overall, this paper presents the development of a light-modulated MXenogel with reversible phase transition using photoresponsive host-guest chemistry. The authors demonstrate the inclusion of both the azobenzene end and alkyl chain spacer of AzoC6 molecules in the cavity of α CD host, resulting in a unique nano-building unit. These units are then tethered on the surfaces of MXene nanosheets and self-assembled into a lamella structure. The microstructure of the MXenogel is characterized using various techniques such as NMR, 2D-ROESY, XPS, WAXS, and SAXS. The MXenogel exhibits reversible photo-responsive

phase transitions upon irradiation with UV or visible light, allowing for facile reprogrammable or reconfigurable charge transportation paths. The authors demonstrate the application of the MXene as a photo-controllable switch in a solid-state circuit. Overall, this work introduces a novel concept and strategy for the creation of MXene-based conductive materials with dynamic functionalities. However, the paper lacks clarity in certain areas, and the authors could provide more detailed explanations of the experimental methods and results. Additionally, the significance and potential impact of the findings are not sufficiently emphasized. Furthermore, the experimental results should be presented with more statistical analysis and replication to strengthen the reliability of the findings. Overall, while the concept is interesting, the paper requires significant revisions to improve its clarity, significance, and robustness.

> We thank the reviewer for recognizing the conceptual idea and the rationality of the material design. In the revised manuscript and Supporting Information, we have modified the contents by taking the comments from the reviewers carefully. More explanations and details regarding the experimental methods have been added to the Results and Discussion section and Supporting Information. In addition, more figures, data, and descriptions, especially on the statistical analyses to strengthen our findings, are also provided.

#Comment 1:

The author demonstrates the reversible isomerization of the AzoC6@2αCD complex through the use of UV and visible light irradiation for 3 and 10 minutes, respectively (Fig. 2g). However, the manuscript lacks a detailed explanation of the differences in intensity between UV and visible light as observed in the UV-vis spectrum (Fig. 2f) and why the visible light condition requires a longer irradiation time. Therefore, it is important to provide a more comprehensive description of these aspects in the manuscript. Specifically, it should explain the absorption characteristics of the AzoC6@2αCD complex under both UV and visible light conditions. The manuscript should clarify whether the observed differences in intensity are due to variations in the absorption cross-sections or the molar absorptivities of the complex for UV and visible light wavelengths. Additionally, the manuscript should discuss any potential shifts in the absorption peaks or changes in the electronic transitions that occur as a result of the isomerization process induced by UV and visible light. Also, please unify units for light condition for UV (mW, mW/cm²) and visible light (lm,

mW/cm²) in method section.

> As suggested by the reviewer, we agree that it is important to provide a more comprehensive description of the absorption characteristics and a detailed explanation of the differences in intensities. As shown in Figure 2f, the maximum at 343 nm and the relatively weak band at 430 nm are corresponding to the transitions of $\pi-\pi^*$ and $n-\pi^*$ of *trans*-AzoC6, respectively. As shown in the newly obtained figure below, the maxima of the AzoC6@2 α CD sample are identical to the pristine AzoC6 in water, indicating that the formation of AzoC6@2 α CD inclusions does not cause the shifting of the absorption bands. After the conversion into the *cis*-form, because of their geometrical differences, the $\pi-\pi^*$ transition blue-shifts to 323 nm and the $n-\pi^*$ slightly red-shifts to 432 nm. After a cycle of UV and visible light irradiations, the *trans*-isomer ratio decreases to 79% and remains at similar values after continuous cyclic irradiations. As for the irradiation time difference between UV and visible lights, according to our previous work, the inclusion formation and the self-assembly of the AzoC6@2 α CD complexes are much slower than the isomerization of Azo moieties. [*Macromolecules*, **2022**, *55*, 8940-8949.] Therefore, a longer irradiation time for the visible light is applied to ensure the successful inclusion formation.

In the revised Supporting Information, we have added the following UV-vis spectra of AzoC6 in water as Figure S4.

Figure S4. UV-vis spectra of AzoC6 in water under different light irradiations.

In the revised manuscript, we have modified the following sentences to discuss more about the UV-vis

spectra.

The maximum at 343 nm and the relatively weak band at 430 nm are corresponding to the transitions of $\pi-\pi^$ and $n-\pi^*$ of trans-AzoC6, respectively. The maxima of the AzoC6@2 α CD sample are identical to the pristine AzoC6 in water (Figure S4), indicating that the formation of AzoC6@2 α CD inclusions does not cause the shifting of the absorption bands. After the conversion into the cis-form, because of their geometrical differences, the $\pi-\pi^*$ transition blue-shifts to 323 nm and the $n-\pi^*$ slightly red-shifts to 432 nm. By irradiating with UV and visible lights, it is shown that the synthesized AzoC6 can reversibly undergo the trans-to-cis and cis-to-trans isomerizations (Figure 2g), respectively. After a cycle of UV and visible light irradiations, the trans-isomer ratio decreases to 79% and remains at similar values after continuous cyclic irradiations.*

In the revised manuscript, we have also added the following sentences to discuss the irradiation time difference.

It should also be noted that, according to our previous work, the inclusion formation and the self-assembly of the AzoC6@2 α CD complexes are much slower than the isomerization of Azo moieties. [Macromolecules 55, 8940-8949 (2022).] Therefore, a longer irradiation time for the visible light is applied to ensure the successful inclusion formation.

We thank the reviewer for pointing out that the units of the light sources in the Methods section should be unified. In the revised manuscript, we have modified the following sentence in the Methods section.

The UV (365-375 nm, 300-700 mW) and visible (5600-6300 K, 700 mW) light sources used in this work were obtained from Vitastar.

#Comment 2:

In the case of the MXene gel depicted in Figure 3, it is important to investigate how the conductivity of the material can vary depending on the concentration with phase condition (solid and gel). The manuscript

should provide a more detailed explanation of this relationship to elucidate the conductivity behavior of the MXene gel with phase status (fig 5d). The manuscript should discuss the experimental methods used to measure the conductivity at different concentrations of the MXene gel. It should highlight the specific concentrations tested and how the conductivity values were obtained. Additionally, the manuscript should present the results graphically, illustrating the relationship between concentration and conductivity.

> As suggested by the reviewer, we have provided the illustration of the electrode configuration and the method for the electrochemical experiments. In brief, a MXene gel is first sandwiched between two ITO glasses and a Teflon spacer is used as a spacer. For a 2-electrode setup, the voltage is applied between the working and the counter electrodes and the current change is probed by an LCR meter. As displayed in Figure S18b, the equivalent circuit is used to fit the obtained raw data of the electrochemical impedance (EIS) spectra. The solution resistance R_s is in series with the parallel combination of a constant phase element (CPE) and an impedance of a faradaic reaction. The charge transfer resistance R_{ct} and the Warburg impedance Z_w are related to the electrochemical and diffusion processes' kinetics, respectively. Figure 5e shows the typical spectra obtained from the MXene gel, in which the Warburg line dominates the impedance plot for a reversible electrochemical system. The suppressed semicircle also implies that the system is dominated by mass transport because of the fast charge-transfer rate.

In the revised Supporting Information, we have added the electrode configuration and equivalent circuit as Figure S19.

Figure S19. (a) Illustration of the electrode configuration for the electrochemical experiments. (b) The equivalent circuit applied in the fitting process.

In the revised Supporting Information, we have also revised the following sentences to discuss the experimental methods used to measure the conductivity at different concentrations of the MXene gel

Figure S19a shows the electrode configuration for the electrochemical measurements. A MXene gel is sandwiched between two ITO glasses, and a Teflon spacer is used as a spacer. In a 2-electrode setup, the voltage is applied between the working and the counter electrodes and the current change is probed by an LCR meter. As displayed in Figure S19b, the equivalent circuit is used to fit the obtained raw data of the electrochemical impedance (EIS) spectra. The solution resistance R_s is in series with the parallel combination of a constant phase element (CPE) and an impedance of a faradaic reaction. The charge transfer resistance R_{ct} and the Warburg impedance Z_w are related to the electrochemical and diffusion processes' kinetics, respectively. Figure 5e shows the typical spectra obtained from the MXene gel, in which the Warburg line dominates the impedance plot for a reversible electrochemical system. The suppressed semicircle also implies that the system is dominated by mass transport because of the fast charge-transfer rate.

To elucidate the conductivity behavior of the MXene gel with phase status, we have proposed two kinds of electron/ion transport paths under different physical conditions: at the gel state, the conduction takes place through the electron transport of the MXene flakes and slower ion transport; at the sol state, because of the increased mobility of the molecules, the conduction takes place through the electron transport of the MXene flakes and faster ion transport.

In the revised Supporting Information, we have added the following proposed mechanism as Figure S21 to explain the electron/ion transport path under different conditions.

Figure S21. Proposed mechanisms of MXene gel conduction under (a) visible (gel state) and (b) UV (sol state) light irradiations. The conductivity of the MXene gel mainly depends on the concentration of the conducting dopants (MXene) and the physical condition of the matrix (AzoC6@2 α CD). Conductions take place through the electron transport of the MXene flakes and the ion migrations.

In the revised manuscript, we have also added the following sentences to discuss the proposed conduction mechanisms.

As shown in Figure S21, we have proposed two kinds of electron/ion transport paths under different physical conditions: at the gel state, the conduction takes place through the electron transport of the MXene flakes and slower ion transport; at the sol state, because of the increased mobility of the molecules, the conduction takes place through the electron transport of the MXene flakes and faster ion transport.

#Comment 3:

In the case of the MXenegel depicted in Figure 3, it is essential for the author to clarify the number of times the phase transition between solidification and liquefaction can be repeated. This information is crucial in assessing the material's durability and long-term applicability. The manuscript should include a comprehensive explanation of the reversibility of the phase transition in the MXenegel. It should specify the number of cycles or repetitions in which the solidification and liquefaction process can be performed without significant degradation or loss of functionality. This could be determined through experimental measurements or by providing an estimation based on the material's stability and previous observations.

> We agree with the reviewer that the material's durability and long-term applicability are crucial. To address this issue, we test a 30 wt% MXenegel with a cyclic electrochemical analysis by irradiating UV and visible lights alternatively. It can be seen that after four full cycles of visible-UV-visible treatment, both the conductivity and the gel-like behavior of the MXenegel are retained. The long-term stability, however, is a challenging issue in terms of the oxidating nature of the $Ti_3C_2T_x$ MXene in an aqueous system. According to previous literature, the Ti atoms in $Ti_3C_2T_x$ MXene may degrade to Ti^{4+} by ca.10 at% after a 14-day storage period in water. [*Adv. Mater. Interfaces* **2022**, 9, 2200480] Because the matrix of the MXenegel is also composed of water, it is possible that the MXenegel can lose its conductive function after a long storage time.

In the revised Supporting Information, we have added the following cyclic test results as Figure S20.

Figure S20. Conductivity analysis of the 30 wt% MXenegel under alternating UV and visible irradiations. The images of the MXenegel at the 0th, 2nd, and 4th cycles are shown.

In the revised manuscript, we have modified the following sentences to mention the cyclic test shown in Figure S20.

To address the crucial information on the material's durability and long-term applicability, we test a 30 wt% MXenegel with a cyclic electrochemical analysis by irradiating UV and visible lights alternatively (Figure S20). After four full cycles of visible-UV-visible treatment, the conductivity and gel-like behavior of the MXenegel are retained.

In the revised manuscript, we have also added the following sentences to discuss the challenges of long-term stability and the possible solution of the MXenegel.

Another major challenge is the long-term stability that arises from the MXene limitations. MXenes are highly sensitive to environmental conditions, particularly to moisture and oxygen. Because the matrix of the MXenegel is composed of water, the MXenegel may oxidize and lose its conductive function after a long period of storage time. According to previous literature, the Ti atoms in $Ti_3C_2T_x$ MXene may degrade to Ti^{4+} by ca.10 at% after 14 days. [Adv. Mater. Interfaces 2022, 9, 2200480] In future work, we will focus on developing a nonaqueous-based MXenegel that can encapsulate MXenes with protective coatings and prevent them from being exposed to oxygen.

Also, in the revised manuscript, we have added the following reference as ref. 47.

47. Zhao, X. et al. The Role of Antioxidant Structure in Mitigating Oxidation in $Ti_3C_2T_x$ and Ti_2CT_x MXenes. *Advanced Materials Interfaces* **9**, 2200480 (2022).

#Comment 4:

The author needs to provide a description of how the mechanical properties of the MXenegel can differ

under UV and visible light irradiation, specifically highlighting how the material is strengthened after the phase transformation. This information is crucial for understanding the impact of light on the mechanical behavior of the MXene/gel. The manuscript should discuss the experimental methods used to evaluate the mechanical properties of the MXene/gel under both UV and visible light irradiation conditions. This may involve techniques such as tensile testing, nanoindentation, or other relevant mechanical characterization methods. The manuscript should also provide details regarding the specific parameters employed during these experiments.

> The reviewer has made a great suggestion about the mechanical properties of the MXene/gel under phase transformation. To evaluate rheological properties, in the revising process, we utilize a rheometer to investigate the MXene/gel under different temperatures in the desired frequency range, as shown in the following figure. Because of the instrument limit, we have conducted the temperature ramping experiments. We use an 8 mm stainless steel plate geometry equipped with a water-circulating temperature-controlled system. From the obtained results, the G' and G'' of the 30 wt% MXene/gel have dropped by nearly 3 orders of magnitude when the temperature increases from 25 to 50 °C. Similar viscosity change can also be observed in the operating shearing range. This is because the self-assembled nanostructure of the AzoC6@2 α CD is mainly driven by the hydrogen bonds between the neighboring α CDs and the hydrophilic heads of the AzoC6 molecules. When the temperature increases above 50 °C, the nanostructure disassembles. The results obtained from the rheological measurements are in good agreement with the DSC results, in which the disassembly temperature at 51 °C is observed.

In the revised Supporting Information, we have added the following rheological data as Figure S10.

Figure S10. (a) Frequency sweep and (b) viscosity shear rate data of the 30 wt% MXene/gel at different temperatures.

In the revised Supporting Information, we have added the following sentences to discuss more the rheological properties of the MXene/gel.

The temperature ramping experiments are conducted in an 8 mm stainless steel plate geometry equipped with a water-circulating temperature-controlled system. The results shown in Figure S10 exhibit the changes of the storage modulus G' and loss modulus G'' and viscosity as a function of frequency oscillating frequency and shear rate, respectively. The G' and G'' of the 30 wt% MXene/gel have dropped by nearly 3 orders of magnitude when the temperature increases from 25 to 50 °C. Similar viscosity change can also be observed in the operating shearing range. This is because the self-assembled nanostructure of the AzoC6@2 α CD is mainly driven by the hydrogen bonds between the neighboring α CDs and the hydrophilic heads of the AzoC6 molecules. When the temperature increases above 50 °C, the nanostructure disassembles. The results obtained from the rheological measurements are in good agreement with the DSC results, in which the disassembly temperature at 51 °C is observed.

Also, in the revised Supporting Information, we have provided details regarding the specific parameters employed during these experiments.

Rheological measurements were collected by a Discovery HR-2 (TA Instruments) equipped with a Peltier plate temperature-controlling system. An 8 mm stainless steel plate geometry was used in our experiments. The axial force for sample compression was set at 1 N and the frequency region was from 0.01 to 100 Hz. The operating temperatures were kept at the desired values with the accuracy controlled to ± 0.2 °C.

#Comment 5:

In the context of conductivity (as depicted in Figures 5 and 6), it is crucial to address the compatibility of thick film with electronic circuit applications. The manuscript should provide a refined statement on this

matter. The manuscript should discuss the implications of thick film conductivity for electronic circuit applications. It should highlight the potential challenges or limitations associated with using thick films in practical electronic circuitry. This could involve considerations such as increased resistance, difficulties in achieving precise patterning, or limitations in miniaturization due to the thickness of the film. Also describe key application of MXenegel for the use of electronic applications.

> The reviewer has raised essential points that the manuscript should highlight the major challenges and emphasize the advantages along with the potential electronic applications of MXenegel.

The MXenegel embedded switch shown in Figure 6 demonstrated that not only MXenegel enables response to different wavelengths of light, but also the reconfigurability and the recyclability inherent to MXenegel platforms allows to reprogram or recycle the circuit when it is damaged. During the recycling procedure, the MXenegel can be liquefied through UV light irradiation and separated from other hard electronics or substrates. After the recycling process, the collected MXenegel can be integrated into another circuit. Such abilities provide a new opportunity to facilitate electronics for sustainable purposes. The amorphous nature of the MXenegel makes it well-suited for soft electronics, which focuses on the development of flexible and stretchable components and circuits.

Despite the advantages brought by combining Azo/ α CD complex and MXene, some potential challenges and limitations still need to be overcome. One major challenge is the environmental tolerance of the MXenegel. Regarding the compatibility of MXenegel in various scenarios, it may require higher temperature resistances and mechanical properties. This could be solved by the structural design of the guest molecules that strengthen the interactions with hosts or increase the inclusion ratio to form a stronger stacking array. Another major challenge is the long-term stability that arises from the MXene limitations. MXenes are highly sensitive to environmental conditions, particularly to moisture and oxygen. Because the matrix of the MXenegel is composed of water, the MXenegel may oxidize and lose its conductive function after a long period of storage time. According to previous literature, the Ti atoms in $Ti_3C_2T_x$ MXene may degrade to Ti^{4+} by ca.10 at% after 14 days. [*Adv. Mater. Interfaces* **2022**, *9*, 2200480] In future work, we will focus on developing a nonaqueous-based MXenegel that can encapsulate MXenes with protective coatings and prevent them from being exposed to oxygen.

In the revised manuscript, we have added the following sentences to provide a more detailed description of the challenges and potential applications of MXenegel.

The MXenegel embedded switch shown in Figure 6 demonstrated that not only MXenegel enables response to different wavelengths of light, but also the reconfigurability and the recyclability inherent to MXenegel platforms allows to reprogram or recycle the circuit when it is damaged. During the recycling procedure, the MXenegel can be liquefied through UV light irradiation and separated from other hard electronics or substrates. After the recycling process, the collected MXenegel can be integrated into another circuit. Such abilities provide a new opportunity to facilitate electronics for sustainable purposes. The amorphous nature of the MXenegel makes it well-suited for soft electronics, which focuses on the development of flexible and stretchable components and circuits.

Despite the advantages brought by combining Azo/ α CD complex and MXene, some potential challenges and limitations still need to be overcome. One major challenge is the environmental tolerance of the MXenegel. Regarding the compatibility of MXenegel in various scenarios, it may require higher temperature resistances and mechanical properties. This issue could be solved by the structural design of the guest molecules that strengthen the interactions with hosts or increase the inclusion ratio to form a stronger stacking array. Another major challenge is the long-term stability that arises from the MXene limitations. MXenes are highly sensitive to environmental conditions, particularly to moisture and oxygen. Because the matrix of the MXenegel is composed of water, the MXenegel may oxidize and lose its conductive function after a long period of storage time. According to previous literature, the Ti atoms in $Ti_3C_2T_x$ MXene may degrade to Ti^{4+} by ca. 10 at% after 14 days. [Adv. Mater. Interfaces 2022, 9, 2200480] In future work, we will focus on developing a nonaqueous-based MXenegel that can encapsulate MXenes with protective coatings and prevent them from being exposed to oxygen.

Also, in the revised manuscript, we have added the following reference as ref. 47.

47. Zhao, X. et al. The Role of Antioxidant Structure in Mitigating Oxidation in $Ti_3C_2T_x$ and Ti_2CT_x MXenes. *Advanced Materials Interfaces* **9**, 2200480 (2022).

Reviewer #3 (Remarks to the Author):

This paper describes the intercalation of azobenzene-containing supramolecular inclusion complexes into MXene nanosheets to form photo-responsive conducting gels. AzoC6@2 α CD inclusion complexes bind to the surface of MXene nanosheets and self-assemble into lamellar structures between MXene layers. Upon irradiation with UV or visible light, MXenegel undergoes a reversible photo-responsive phase transition. The application of MXenegel to solid-state circuits and to optical controllable switches is very interesting. However, MXenegel's explanation of reversible phase transitions is insufficient, and the reviewer would claim major revision. Specific points are shown below:

> We appreciate the reviewer for pointing out the lack of critical explanations of the MXenegel mechanism. In the revising process, we have carefully read the comments from the reviewers and made a number of modifications to the manuscript and Supporting Information. For major points, additional experiments, including XPS, WAXS, zeta potential, and chemical simulations, are conducted; for minor points, we have also made several changes and additions by taking the advice of the reviewer.

There are questions about the mechanism of the reversible photo-responsive phase transition in MXenegel. As shown in Figure 3c, the Azo@2 α CD gel gradually changes to a sol-like structure when irradiated with UV light. The photo-responsive behavior of MXenegel is also identical, as shown in Figure 3d. In the case of Figure 3d, this may be due to the properties of free AzoC6@2 α CD, which is not intercalated into the MXene nanosheet. For this to be the case, the change in thickness of the MXene nanosheet before and after UV irradiation must be clearly shown.

> The reviewer has pointed out an important issue regarding the phase behaviors of the Azo@2 α CD and the MXenegel. The macroscopic investigation of the sol-gel transition of the MXenegel and Azo@2 α CD is identical, bringing a critical question as to whether the photo-responsive behavior of MXenegel is due to the properties of free AzoC6@2 α CD. To address the question raised by the reviewer, it is important to study two main issues: 1) Can the AzoC6 molecules be able to intercalated in the MXene layers in the gel matrix? 2) If so, are the intercalated inclusions able to undergo the photoresponsive assemble/disassemble behaviors?

For the first issue, we have conducted the XPS analyses to investigate the interactions between the AzoC6 molecules and MXene flakes. The samples are prepared in the aqueous state, filtered by a 200 nm filter paper and washed with DI water several times to remove the unattached AzoC6 molecules. The MXene/AzoC6 XPS results clearly show that the surfaces of the MXene flakes are attached by plenty of AzoC6 molecules, as evidenced by the formation of the N1s peak, the C-N peak in the C1s spectrum, and the C-O peak in the O1s spectrum; for the MXene gel sample with a high MXene concentration (30 wt%), the strong C-C in C1s, the C-O peak in O1s, and the N 1s peak, indicate the successful intercalation of AzoC6@2 α CD inclusions. The results clearly show the successful interactions even after several DI water rinsing processes. In addition, it can be confirmed from the high-resolution scan of the Ti 2p spectra that there is no noticeable oxidation of the MXene nanosheets occurring, as the TiO₂ peak remains weak in all of the samples.

In the revised Supporting Information, we have added the newly obtained XPS data as Figures S12 and S13. The fitting results are also shown in Tables S1 and S2.

Figure S12. XPS component peak fits of MXene: (a) Ti 2p, (b) C 1s, (c) F 1s, (d) O 1s, and (e) N 1s. The sample is prepared in the aqueous state, filtered by a 200 nm filter paper, and washed with DI water several times.

Table S1. MXene ($\text{Ti}_3\text{C}_2\text{T}_x$) XPS peak fitting results

Element	Element AT%	Binding energy (eV)	Component	Component AT%	FWHM
Ti 2p_{3/2} (2p_{1/2})	21.1	455.2 (461.2)	Ti-C	20.5	0.9 (1.9)
		455.9 (461.7)	Ti2+	33.6	1.4 (1.3)
		457.0 (462.8)	Ti3+	40.1	2.1 (1.7)
		459.2 (464.1)	TiO ₂	5.8	1.7 (1.3)
C 1s	49.5	282.3	C-Ti-Tx	48.0	1.0
		284.8	C-C	47.6	2.3
		285.7	CH _x /CO	4.4	2.6
O 1s	20.3	530.0	TiO ₂	55.3	1.2
		531.1	C-Ti-O _x	13.6	1.1
		532.1	C-Ti-(OH) _x	21.6	1.6
		533.5	H ₂ O	9.5	1.7
F 1s	9.1	687.5	C-Ti-F _x	4.6	1.9
		685.0	TiO ₂ -x _F	76.8%	1.4
		686.1	Al-F	18.6%	1.4

Figure S13. XPS component peak fits of MXene/AzoC6: (a) Ti 2p, (b) C 1s, (c) F 1s, (d) O 1s, and (e) N 1s. The sample is prepared in the aqueous state, filtered by a 200 nm filter paper, and washed with DI water several times to remove the unattached AzoC6 molecules.

Table S2. MXene/AzoC6 XPS peak fitting results

Element	Element AT%	Binding energy (eV)	Component	Component AT%	FWHM
Ti 2p _{3/2} (2p _{1/2})	21.2	455.4 (461.1)	Ti-C	20.8	1.1 (1.3)
		456.4 (462.0)	Ti ²⁺	54.0	2.0 (1.7)
		458.0 (463.4)	Ti ³⁺	18.4	1.9 (1.9)

		459.6 (464.0)	TiO ₂	6.8	1.6 (4.0)
C 1s	51.9	282.3	C-Ti-T _x	19.6	1.1
		284.8	C-C	28.2	2.3
		286.6	CH _x /CO	34.0	1.9
		285.3	C-N	18.2	1.3
O 1s	13.0	529.9	TiO ₂	42.2	1.3
		531.0	C-Ti-O _x	16.5	1.4
		532.0	C-Ti-(OH) _x	13.9	2.4
		533.5	C-O	14.0	2.0
		532.8	H ₂ O	13.4	1.7
F 1s	8.5	685.0	C-Ti-F _x	86.2	2.1
		685.6	TiO _{2-x} F _x	11.0	2.5
		688.0	Al-F	2.9	2.0
N 1s	5.4	399.2	N=N/IM-N	37.7	2.0
		401.5	IM-N ⁺	62.3	1.5

In the revised manuscript, we have modified the following sentences to discuss the newly obtained XPS data.

High-resolution XPS element scans of the MXene, MXene/AzoC6, and MXenegel are shown in Figures S12, S13, and S14, respectively. For the MXene/AzoC6 sample, even after being washed with DI water several times, the results clearly show that the surfaces of the MXene flakes are attached by plenty of the AzoC6 molecules, as evidenced by the formation of the N1s peak, the C-N peak in the C1s spectrum, and

the C-O peak in the O1s spectrum; for the MXene/gel sample, the strong C-C in C1s, the C-O peak in O1s, and the N 1s peak, indicate the successful intercalation of the AzoC6@2 α CD inclusions. In addition, it can be confirmed from the high-resolution scan of the Ti 2p spectra that there is no noticeable oxidation of the MXene nanosheets occurring, as the TiO₂ peak remains weak in all of the samples.

Also, in the revised Supporting Information, we have added the following sentences to give the details of the sample preparation.

For MXene and MXene/AzoC6 XPS spectra, the samples were prepared in the aqueous state, filtered by a 200 nm filter paper, and washed with DI water several times to remove any possible residue and the unattached AzoC6 molecules; for the MXene/gel spectra, the samples were prepared by the vacuum-dried method.

As for the second issue, to discuss whether the intercalated inclusions can undergo photoresponsive assemble/disassemble behaviors, we apply SAXS analyses. It should be noted that the assembly/disassembly of the AzoC6@2 α CD bilayers can only be realized in the presence of water molecules. The quality of the WAXS signals of the non-drying MXene/gel, however, is poor. Thus, instead of investigating the layer thickness change using WAXS, we conduct the SAXS analyses of the highly doped MXene/gel samples under different conditions. As shown in the newly obtained SAXS results, the maxima disappear after UV irradiation/thermal treatment, indicating the disassembly of the intercalating AzoC6@2 α CD bilayers. Because the MXene concentration of the MXene/gel is as high as 30 wt%, we conclude that the photo- and thermal responsive behaviors mainly originate from the AzoC6@2 α CD intercalated between the MXene nanosheets.

In the revised Supporting Information, we have added the following SAXS spectra as Figure S18.

Figure S18. SAXS profiles of (a) the 30 wt% MXene gel and (b) after UV and (c) after thermal treatment (50 °C).

Also, in the revised manuscript, we have modified the following sentences to discuss the photoresponsive behavior of the intercalating inclusions in the MXene gel in the presence of water.

It should be noted that the assembly/disassembly of the AzoC6@2 α CD bilayers can only be realized in the presence of water molecules. As shown in Figure S17, the SAXS profile of the vacuum-dried MXene gel exhibits the smallest d-spacing value of 2.07 nm, indicating the collapsing of the bilayer structure. This result is in line with the WAXS analyses that fast water evaporation can induce the destruction of Azo@2 α CD bilayers between the MXene. As shown in Figure S18, the maxima disappear after UV irradiation/thermal treatment, indicating the disassembly of the intercalating AzoC6@2 α CD bilayers. Because the MXene concentration of the MXene gel is as high as 30 wt%, we conclude that the photo- and thermal responsive behaviors mainly originate from the AzoC6@2 α CD intercalated between the MXene nanosheets.

In terms of supramolecular chemistry, it should explain why the bilayer structure of the AzoC6@2 α CD

complex is broken when Azo@2 α CD is changed to Azo@ α CD by UV irradiation.

> According to previous research, the self-assembly of the small-molecule/cyclodextrin system mainly depends on the chemical structure of the small molecule and the host/guest ratio. [*Soft Matter*, **2011**, *7*, 10417-10423] To give a deeper understanding of the host/guest ratio of the inclusions, we calculate the optimized geometry of the cis- and trans- AzoC6 by Gaussian using the B3LYP/6-31G (d, p) method. According to the results, the length of the trans-AzoC6 is ca. 2.18 nm, in which the hydrophobic part (alkyl chain and trans-azobenzene) is ca. 1.78 nm. The length of two and three α CD molecules is ca. 1.58 and 2.37 nm, respectively, indicating that one trans-AzoC6 guest can be included in a maximum of two α CD hosts. The cis-form AzoC6 molecule, however, only has ca. 0.74 nm of alkyl chain that can be included in one α CD molecule because the size and polarity of the cis-azobenzene are not fit to the α CD cavity. Based on the scattering and simulating results, we deduce that the disassembly of the bilayer structure is triggered by the weaker hydrogen bonding between the AzoC6@ α CD inclusions, which cannot stack into a lamellae structure, compared with the AzoC6@2 α CD.

In the revised Supporting Information, we have added the following simulation results as Figure S8.

Figure S2. B3LYP/6-31G (d, p)-optimized structure of a cis- and trans- AzoC6.

In the revised manuscript, we have added the following sentence to mention the inclusion formation and the related reference.

According to previous research, the self-assembly of the small-molecule/cyclodextrin system mainly depends on the chemical structure of the small molecule and the host/guest ratio. Thus, the ratio between AzoC6 and α CD plays a critical role in this work (Figure S8). [Soft Matter 7, 10417-10423 (2011)]

In the revised manuscript, we have added the following reference as ref. 43.

43. Jiang, L., Yan, Y. & Huang, J. Zwitterionic surfactant/cyclodextrin hydrogel: microtubes and multiple responses. *Soft Matter* 7, 10417-10423 (2011).

Also, in the revised Supporting Information, we have added the following sentences to describe the findings of our simulated models and give more descriptions regarding the self-assembly of the AzoC6/ α CD inclusions.

To elucidate the host/guest ratio of the inclusions, the optimized geometry of the cis- and trans- AzoC6 is calculated by Gaussian using the B3LYP/6-31G (d, p) method. As shown in Figure S8b, the length of the trans-AzoC6 is ca. 2.18 nm, in which the hydrophobic part (alkyl chain and trans-azobenzene) is ca.

1.78 nm. The length of two and three α CD molecules is ca. 1.58 and 2.37 nm, respectively, indicating that one trans-AzoC6 guest can be included in a maximum of two α CD hosts. The cis-form AzoC6 molecule shown in Figure S8a, however, only has ca. 0.74 nm of alky chain that can be included in one α CD molecule because the size and polarity of the cis-azobenzene are not compatible with α CD cavity. Based on the scattering and simulating results, we deduce that the disassembly of the bilayer structure is triggered by the weakening of the hydrogen bonding between the AzoC6@ α CD inclusions, which cannot stack into a lamellae structure, compared with the AzoC6@2 α CD.

We appreciate all the constructive feedback from the reviewers and hope the manuscript is now suitable for publication in *Nature Communications*.

REVIEWERS' COMMENTS

Reviewer #2 (Remarks to the Author):

I would like to express my gratitude to you for diligently addressing the concerns raised during the review process. I particularly appreciate the effort you took to enhance the manuscript by incorporating more detailed explanations, figures, data, and additional descriptions in the Results and Discussion section, as well as in the Supporting Information. The inclusion of statistical analyses to validate your findings further augments the credibility of your research.

Based on these comprehensive modifications and the clarity with which you have presented the conceptual ideas and material design rationale, I believe that the manuscript is now in a robust shape.

Consequently, I am of the opinion that this manuscript is now suitable for acceptance. I look forward to seeing it published and believe it will make a valuable contribution to the field.

=====

In reviewing the authors' responses to the comments of reviewer #1, it is evident that they have thoroughly and effectively addressed each point raised. The inclusion of specific temperature behavior, the addition of high-resolution SEM and TEM images, and the detailed explanation of light-responsive phase behavior with supporting FTIR spectra significantly enhance the manuscript's clarity and scientific rigor.

The authors have also commendably expanded the discussion on the storage stability and rheological properties of MXene, providing valuable insights into its nature and behavior. Furthermore, the new data on the dose-dependent effects of MXene on MXene's architecture and the inclusion of experiments demonstrating the recyclability of electronic components substantiate their claims and add considerable depth to their research. Lastly, the effort to define all abbreviations at their first occurrence and improve the standardization and aesthetics of the figures demonstrates meticulous attention to detail. Overall, the authors have provided comprehensive and satisfactory responses to all the queries and suggestions, significantly enriching the quality and impact of their work. The manuscript is now proper to be accepted in Nature Communications after responding to some minor comments as below.

p.s., For minor comment,

0. It will be much better to change the schematic image of Figure 1 because Azo C6 in MXene was not well expressed.

1. There are some redundant outer lines near Figure 2(e), (f), (h), and, the gray color in 2(g) should be removed with a larger symbol size

2. Similarly, in Figure 4(e), and (f) some outer lines should be removed

3. Also, in Figure 5a, there are black rectangular (please remove)

4. 5d, e, all dots and notation in x and y-axis are small, please change

=====

Reviewer #3 (Remarks to the Author):

The manuscript has been revised carefully according to the reviewers' comments. The added phrases in the text, analytical data in the Supporting Information, and the reference support the claim. I would very much like to see it published in Nature Communications.

Point-by-point response to the reviewers' comments

REVIEWERS' COMMENTS

Reviewer #2 (Remarks to the Author):

I would like to express my gratitude to you for diligently addressing the concerns raised during the review process. I particularly appreciate the effort you took to enhance the manuscript by incorporating more detailed explanations, figures, data, and additional descriptions in the Results and Discussion section, as well as in the Supporting Information. The inclusion of statistical analyses to validate your findings further augments the credibility of your research.

Based on these comprehensive modifications and the clarity with which you have presented the conceptual ideas and material design rationale, I believe that the manuscript is now in a robust shape.

Consequently, I am of the opinion that this manuscript is now suitable for acceptance. I look forward to seeing it published and believe it will make a valuable contribution to the field.

> We appreciate the reviewer for recognizing the effort we made in the revision process.

=====
In reviewing the authors' responses to the comments of reviewer #1, it is evident that they have thoroughly and effectively addressed each point raised. The inclusion of specific temperature behavior, the addition of high-resolution SEM and TEM images, and the detailed explanation of light-responsive phase behavior

with supporting FTIR spectra significantly enhance the manuscript's clarity and scientific rigor.

The authors have also commendably expanded the discussion on the storage stability and rheological properties of MXene/gel, providing valuable insights into its nature and behavior.

Furthermore, the new data on the dose-dependent effects of MXene on MXene/gel's architecture and the inclusion of experiments demonstrating the recyclability of electronic components substantiate their claims and add considerable depth to their research.

Lastly, the effort to define all abbreviations at their first occurrence and improve the standardization and aesthetics of the figures demonstrates meticulous attention to detail. Overall, the authors have provided comprehensive and satisfactory responses to all the queries and suggestions, significantly enriching the quality and impact of their work.

The manuscript is now proper to be accepted in Nature Communications after responding to some minor comments as below.

p.s., For minor comment,

0. It will be much better to change the schematic image of Figure 1 because Azo C6 in MXene/gel was not well expressed.

1. There are some redundant outer lines near Figure 2(e), (f), (h), and, the gray color in 2(g) should be removed with a larger symbol size

2. Similarly, in Figure 4(e), and (f) some outer lines should be removed

3. Also, in Figure 5a, there are black rectangular (please remove)

4. 5d, e, all dots and notation in x and y-axis are small, please change

> We have made several changes/modifications according to the reviewer and the guidance from the Editorial Office. The outer lines and gray shadows are removed. All dots and notations in the x and y-axis

are clearly shown. We have also provided the vector-based .ai file for all the figures shown in the manuscript files. We believe that the quality of the figures is now substantially improved.

=====

Reviewer #3 (Remarks to the Author):

The manuscript has been revised carefully according to the reviewers' comments. The added phrases in the text, analytical data in the Supporting Information, and the reference support the claim. I would very much like to see it published in Nature Communications.

> We thank the reviewer's positive comments.

We appreciate all the constructive feedback from the reviewers and hope the manuscript is now suitable for publication in *Nature Communications*.